# Black Sheep in the Herd: Playing with Spuriously Correlated Attributes for Vision-Language Recognition

**Xinyu Tian**[1], **Shu Zou**[1], **Zhaoyuan Yang**[2], **Mengqi He**[1], **Jing Zhang**[1]
[1]Australian National University,    [2]GE Research

## Abstract

Few-shot adaptation for Vision-Language Models (VLMs) presents a dilemma: balancing in-distribution accuracy with out-of-distribution generalization. Recent research has utilized low-level concepts such as visual attributes to enhance generalization. However, this study reveals that VLMs overly rely on a small subset of attributes on decision-making, which co-occur with the category but are not inherently part of it, termed spuriously correlated attributes. This biased nature of VLMs results in poor generalization. To address this, 1) we first propose Spurious Attribute Probing (SAP), identifying and filtering out these problematic attributes to significantly enhance the generalization of existing attribute-based methods; 2) We introduce Spurious Attribute Shielding (SAS), a plug-and-play module that mitigates the influence of these attributes, seamlessly integrating into various Parameter-Efficient Fine-Tuning (PEFT) methods. In experiments, SAP and SAS significantly enhance accuracy on distribution shifts across 11 datasets and 3 generalization tasks while preserving downstream performance, establishing a new state-of-the-art benchmark. The code will be available here.

## 1 Introduction

The emergence of large-scale pre-trained Vision-Language Models (VLMs) (Radford et al., 2021; Li et al., 2022a) bridges the gap between images and texts. However, conventional fine-tuning of these models entails significant computational burdens, leading to Parameter-Efficient Fine-Tuning (PEFT), such as prompt tuning (Khattak et al., 2023a; Zhou et al., 2022a), adapters (Sung et al., 2022; Gao et al., 2024) and LoRA (Hu et al., 2021; Dettmers et al., 2024). With PEFT, requiring approximately 1% of model parameters, one may adeptly adapt to downstream tasks, achieving comparable or even superior performance to full fine-tuning (Liu et al., 2021). Yet, recent studies have revealed that in few-shot scenarios where observed samples are limited, PEFT struggles to generalize to out-of-distribution datasets and may compromise the VLMs' strong zero-shot capability (Zhou et al., 2022b; Yao et al., 2023; Bulat & Tzimiropoulos, 2023). This creates a trade-off where individuals aim for strong performance on downstream tasks while endeavoring to maintain the ability of VLMs to handle distribution shifts.

In response to the above-mentioned challenges, various strategies have been proposed such as category conditioning (Zhou et al., 2022b; Yao et al., 2024), prompt regularization (Yao et al., 2023; Bulat & Tzimiropoulos, 2023; Khattak et al., 2023b) and training-free adaptation (Udandarao et al., 2022; Zhang et al., 2022b). Recently, it has been discovered that incorporating descriptors, also known as visual attributes during training can significantly improve the accuracy of adapted modules on out-of-distribution datasets (Zhang et al., 2024c; Liao et al., 2024; Tian et al., 2024; Ma et al., 2023; Liu et al., 2024b). The motivation behind these works is that attributes, as lower-level concepts, are more likely to establish connections to unseen categories compared to the high-level names (Zhang et al., 2024c; Tian et al., 2024). These methods can be divided into two types: one involves generating visual attributes for the target category using Large Language Models (LLMs) (Tian et al., 2024; Ma et al., 2023; Liu et al., 2024b), while the other entails searching for optimal attributes from a pre-defined vocabulary that maximizes semantic similarity (Zhang et al., 2024c) or training accuracy (Liao et al., 2024). Yet, a commonality among them is their reliance on the set of generated attributes, *i.e.*, the attribute pool.

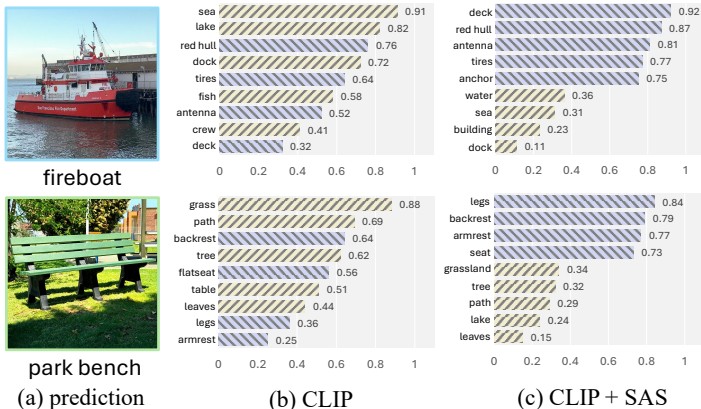

(a) prediction      (b) CLIP      (c) CLIP + SAS

Figure 1: **The phenomenon of Black Sheep in the Herd**. We rank attribute weights on VLM predictions using CBMs, with yellow and purple bars to denote spurious and core attributes respectively. In **(b)**, we observe that for vanilla VLMs, 2 out of the top-3 are spurious attributes, heavily influencing decisions. In **(c)**, SAS mitigates this by suppressing the influence of spurious attributes.

While promising, the limitations of this line of work have been underexplored. Initial suspicions emerge from Roth et al. (2023), which find that in certain cases, replacing attributes with random sequences does not lead to a notable performance decline. Subsequently, An et al. (2023) discover that VLMs sometimes disregard the presence of attributes, leading to minimal gains. This prompts us to inquire: *Are attributes truly dependable? If so, then whence do these failure cases arise?*

To tackle the aforementioned inquiries, we conduct a manual examination of the attribute pool generated by existing methods. We stumble upon an often overlooked fact: while most attributes accurately depict the intrinsic characteristics of the target category, there exists a small subset of attributes that co-occur with the category but are not part of it, leading to strong spurious correlations. For instance, when querying LLM with what does a mountain bike look like we receive attributes like wheels, handle, and basket, yet unexpected attributes such as trees and road also emerge. This phenomenon is also observed in vocabulary-based methods, where attributes are chosen based on in-distribution samples. Inspired by Singla & Feizi (2021), we refer to the former as *core attributes* and the latter as *spuriously correlated attributes*[1].

Building upon the motivation to enhance generalization, a natural idea is to strive for a "pure" attribute pool that accurately reflects the true characteristics of the category. Therefore, we conduct a simple experimental study based on existing methods (Zhang et al., 2024c; Tian et al., 2024) where we manually identify attributes that might lead to spurious correlations with the target category and remove them. Despite the small proportion of these attributes ($< 7\%$), we observe a significant improvement in out-of-distribution generalization accuracy. To gain deeper insights into how spurious attributes affect VLMs, we employ concept bottleneck models (CBMs) (Koh et al., 2020; Yang et al., 2023), a well-established method for interpreting and ranking attribute weights in decision-making processes. Our analysis reveals that despite their small representation in the overall pool, spurious attributes exert a significant influence, particularly among the top-3 attributes on decision-making as depicted in Fig. 1. We term this phenomenon *Black Sheep in the Herd* since 1) spurious attributes act as the "black sheep" within the pool, constituting a small fraction; 2) nevertheless, this small fraction significantly impacts the generalization ability of VLMs.

We could utilize the aforementioned manual inspection to assist existing attribute-based methods. However, this remains a mere fancy dream since manually identifying spurious attributes in the pool is prohibitively expensive. This motivates us to devise a new method for generating a pure pool, one that contains only those core attributes belonging to the category. Hence, we propose SPURI-OUS ATTRIBUTE PROBING (SAP), an approach to derive an attribute pool where core and spurious attributes are clearly separated. SAP integrates Multi-modal Large Language Models (MLLMs) and Concept Bottleneck Models (CBMs) to tackle this challenge. Leveraging MLLMs, SAP initially distinguishes core attributes from non-core counterparts, and then CBMs prioritize the latter by selecting those with a significant impact on model decisions as spurious attributes. SAP complements existing attribute-based methods and, to the best of our knowledge, is the first approach to identifying spurious attributes in open-world settings without explicit human supervision.

Despite the effectiveness of SAP, it faces limitations: It may prevent the presence of spurious attributes in the language branch, yet it cannot stop the model from learning spurious features. This ex-

---

[1] We refer spuriously correlated attributes to spurious attributes for brevity.

tends the scope beyond attribute-based methods, and will result in poor generalization across PEFT family. Therefore, we propose SPURIOUS ATTRIBUTE SHIELDING (SAS), a module to mitigate the influence of spurious features which can be seamlessly integrated into arbitrary PEFT methods. Specifically, SAS introduces a subsidiary task by creating a set of pseudo categories defined by spurious attributes alongside the real ones, allowing VLMs to distinguish between them. For instance, if streetlight is considered a spurious attribute for the target category vehicle, we establish a separate pseudo category exclusively for streetlight and discern between the two, thus decreasing the dependency on streetlight for identifying vehicle. The experiments show that by combining SAS into existing PEFT approaches, the accuracy under distribution shifts is significantly improved, reaching a new state of the art.

In summary, our main contributions are as follows:

- Despite the promise of visual attributes in various applications, we discover a group of *black sheep*, *i.e.*, spurious attributes, on which VLMs inherently heavily rely, thereby leading to poor generalization and robustness.

- We introduce SPURIOUS ATTRIBUTE PROBING (SAP), aiming to identify and eliminate these problematic attributes, thereby substantially improving the generalization of current attribute-based methods.

- We present SPURIOUS ATTRIBUTE SHIELDING (SAS), a plug-and-play module seamlessly integrating into various PEFT methods to mitigate the influence of spurious attributes on predictions.

## 2 RELATED WORK

**Vision-Language Models.** Recently, it has been discovered that associating text and images for pre-training, instead of using images alone, enables powerful zero-shot capability, leading to VLMs. Initially, simple dual-tower structures are employed, where the representations of the two modalities are modeled by separate encoders and connected via contrastive learning, *i.e.*, CLIP (Radford et al., 2021). Subsequently, more works have been built upon this foundation. For instance, Li et al. (2022a) bridge two encoders by fusion for better cross-modality interactions, Li et al. (2023b) employ masked image modeling to achieve a trade-off between accuracy and training time, and Li et al. (2022b) incorporate visual detection and grounding in pre-training for object-level reasoning. For more information, we refer to Zhang et al. (2024a) for a detailed survey of recent VLMs.

**Parameter-Efficient Fine-Tuning.** As pre-trained models grow larger, traditional fine-tuning demands significant resources, leading to PEFT (Hu et al., 2021; Liu et al., 2021; Lester et al., 2021; Houlsby et al., 2019). However, PEFT is a double-edged sword: while it adeptly adjusts to downstream tasks, it also brings poor generalization to the open world, inspiring various current remedies. For instance, category conditioning (Zhou et al., 2022b; Yao et al., 2024) infuses category-aware knowledge for discriminative and generalizable learning, prompt regularization (Yao et al., 2023; Bulat & Tzimiropoulos, 2023) confines learnable prompts to corresponding textual features, and training-free adaptation (Udandarao et al., 2022; Zhang et al., 2022b) eschews gradient-based optimization to prevent overfitting. Recently, another line of work leveraging visual attributes has shown promising results, achieving state-of-the-art performance in various generalization tasks.

**Visual Attributes for Recognition.** The initial exploration of attributes for recognition begins in zero-shot settings (Menon & Vondrick, 2023; Pratt et al., 2023), where individuals utilize attributes generated by LLMs to offer more expressive and accurate descriptions. Subsequently, it is observed that training VLMs to grasp fundamental concepts like visual attributes aids in generalizing to unseen data, prompting a surge in attribute-based methods. For instance, Tian et al. (2024) appends attributes to category names, Liao et al. (2024) initializes learnable tokens as attribute embeddings, and Ma et al. (2023) adopts a more aggressive approach by replacing category names entirely with attributes. Additionally, Wei et al. (2019) utilize adversarial training to learn attribute-object composition, while Huang et al. (2024) and Wang et al. (2015) improve the model's fine-grained understanding by building multi-granularity and hierarchical attributes. Nonetheless, recent studies have noted a decline in attribute effectiveness in certain scenarios (An et al., 2023), sometimes reducing to a mere ensembling effect (Roth et al., 2023). This paper delves into the issue, attributing it to spurious attributes, and proposes two plug-and-play approaches to complement existing methods.

**Spurious Attribute Identification.** Spurious attributes arise from model debiasing (Seth et al., 2023; Chuang et al., 2023; Berg et al., 2022), defined as *those likely to co-occur with the object but not part of it* (Singla & Feizi, 2021). Although a well-known term, it remains underexplored due to the difficulty in identification. The initial endeavor by Singla & Feizi (2021) involves manually labeling to identify spurious attributes. Similarly, Wong et al. (2021) integrates human supervision with sparse linear layers to mitigate labor expenses. Others identify spurious attributes by analyzing their properties. For instance, Wu et al. (2023) observe that spurious attributes exhibit instability across data environments and introduce concept sensitivity for identification. Conversely, Teotia et al. (2022) train an attribute probing network to predict spurious attributes. Recently, the work most akin to ours, Adila et al. (2023), utilize LLMs to derive harmful insight representations by comparing differences between concepts. However, the inference complexity of this method escalates exponentially with the number of concepts, restricting its application to small-scale datasets. In contrast, our proposed `SAP` 1) necessitates neither human labeling nor a training process, rendering it extremely cost-effective; and 2) is scalable to any large-scale dataset, *e.g.*, ImageNet.

**Spurious Correlation Mitigation.** Current spurious mitigation methods can be mainly categorized into two types. The first assumes that spurious attributes within the dataset are either unknown or complex, employing various proxies to mitigate spurious correlations. For instance, Xu et al. (2020); Yao et al. (2022); Han et al. (2022) introduce augmentation via domain mix-up to learn invariant features, while Li et al. (2022c); Zhang et al. (2022a); Utama et al. (2020) advocate for instance reweighting to emphasize hard samples. Others calibrate biased representation through contrastive learning (You et al., 2024; Zhang & Ré, 2022). The second type explicitly assumes that spurious correlations arise from known attributes (Chuang et al., 2023; Berg et al., 2022). For instance, Wu et al. (2023) balance training data by swapping spurious concepts among categories, whereas Adila et al. (2023) calibrate embeddings by removing spurious representations. In contrast, `SAS` belongs to the latter, where the attribute prior is known, and thanks to the effectiveness of `SAP`, it may accurately mitigate spurious correlations caused by identified spurious attributes. For further details, a quantitative comparison between `SAS` and related works is provided in Supp. Mat. B.

## 3 METHOD

### 3.1 PROBLEM SETUP

We assume the training set of pairs $\mathcal{D} = \{(x, c)\}$, where $x \in \mathcal{X}$ and $c \in \mathcal{C}$ represent the image and ground truth label, respectively. The attribute-based methods aim to construct a category-wise prompt $t_c = f(c, \mathcal{P})$ such that the conditional distribution of the prediction $y$ given $x$ is modeled as

$$P(y|x) = \frac{\exp(s(\phi_I(x), \phi_L(t_y))/\tau)}{\sum\limits_{c \in \mathcal{C}} \exp(s(\phi_I(x), \phi_L(t_c))/\tau)}, \tag{1}$$

where $\phi_I$ and $\phi_L$ represent the vision and language encoder, respectively, $s(\cdot, \cdot)$ indicates the similarity function, and $\tau$ is the temperature scaler. $\mathcal{P} = \{\mathcal{A}_c \mid c \in \mathcal{C}\}$ is an attribute pool generated by $\mathcal{A}_c = \mathcal{U}(\mathcal{H}(c))$ such that $\mathcal{A}_c = \{a_c^1, a_c^2, ..., a_c^J\}$. Depending on previous work, $\mathcal{U}$ could be a LLM, thus $\mathcal{H}(c)$ is a set of LLM prompts incorporating the category name of $c$. $\mathcal{U}$ could also be a large vocabulary, such that $\mathcal{H}(c)$ becomes a key to search for the semantically related attributes. Therefore, $f(\cdot, \cdot)$ is denoted as a concatenation function to integrate the category name and corresponding attributes together. The optimization objective is typically a cross-entropy loss $\mathcal{L}_{ce}$.

It's important to mention that in this work, we refrain from specifying particular learnable parameters; they could encompass learnable prompts, adapters, or LoRA. Our goal is to ensure the versatility of our plug-and-play method across various PEFT approaches.

### 3.2 MOTIVATION

We present the motivation of this work by revealing an overlooked fact: the attribute pool in current methods are not purely aligned with the intrinsic semantics of categories. Some attributes are spurious, co-occurring with categories but not inherently linked to them. To investigate the impact of these "black sheep", we conduct a simple experimental study. Specifically, we manually traverse the attribute pool $\mathcal{P}$ across various methods and identify spurious attributes within. We use a conventional method following Singla & Feizi (2021) with a simplistic version. Given the category $c$,

| Method | FGVCAircraft | | | SUN397 | | | Flowers102 | | | DTD | | | Average | | |
|---|---|---|---|---|---|---|---|---|---|---|---|---|---|---|---|
| | Base | New | SR | Base | New | SR | Base | New | SR | Base | New | SR | Base | New | SR |
| CPL | 42.27 | 38.85 | 5.43 | 81.88 | 79.65 | 6.61 | 98.07 | 80.43 | 5.71 | 80.92 | 62.27 | 5.13 | 75.79 | 65.30 | 5.72 |
| CPL - SA | **42.62** | 41.33 | — | 82.14 | **82.36** | — | 98.35 | **82.16** | — | **81.62** | 64.77 | — | **76.18** | 67.66 | — |
| ArGue | 41.29 | 38.80 | 5.13 | 81.89 | 80.48 | 6.45 | 98.62 | 77.96 | 6.69 | 80.33 | 67.03 | 5.97 | 75.53 | 66.07 | 6.06 |
| ArGue* | 41.34 | 39.34 | 4.86 | 81.97 | 80.62 | 6.11 | 98.58 | 78.11 | 6.44 | 80.41 | 67.26 | 5.62 | 75.58 | 66.33 | 5.76 |
| ArGue - SA | 41.55 | **41.60** | — | **82.33** | 81.94 | — | **98.73** | 78.75 | — | 80.79 | **68.47** | — | 75.85 | **67.69** | — |

Table 1: **The results on base-to-new generalization before and after removing spurious attributes (SA) from the pool.** We report accuracy on base and new categories, and spurious rate (SR), which refers to the proportion of spurious attributes to the entire pool.

we randomly sample 5 images from the shots and visualize the heatmap. For specific attribute $a_c^k$, we determine whether it is a part of the main object, or separate objects in the background based on the sampled images with the heatmap activations.

Upon identifying these spurious attributes, we remove them from the pool and compare the changes in their generalization capability before and after elimination. The experiment is evaluated on base-to-new generalization, following the outlined settings in Section 4. As baselines, we select CPL (Zhang et al., 2024c) and ArGue (Tian et al., 2024), representing vocabulary-based and LLM-assisted methods, respectively. Additionally, we consider a variant, ArGue*, where we modify the LLM prompts to reduce the likelihood of spurious attribute occurrence. For instance, we append an additional instruction focus on mountain bike itself to the original prompt. Further details are provided in Supp. Mat. A.

**Removing spurious attributes significantly enhances generalization.** While most attributes contribute positively to generalization, spurious attributes are exceptions to this trend. Removing these exceptions leads to a notable increase in accuracy on the new category on average (65.30% → 67.66% for CPL and 66.07% → 67.69% for ArGue), without compromising accuracy on the base category. This phenomenon is aptly described as Black Sheep in the Herd since 1) spurious attributes constitute only a small portion of the pool (< 7%); 2) yet this small portion significantly impacts the generalization ability of VLMs.

**VLMs heavily rely on spurious attributes for predictions.** To deepen our understanding of this phenomenon, we use concept bottleneck models (CBMs) (Koh et al., 2020) to determine attribute weights in model decision-making. In Fig. 1, attributes are ranked by weight from high to low. Among the top-3 attributes influencing predictions, spurious attributes occupy two positions. For instance, in predicting fireboat, VLMs heavily rely on sea and lake as crucial concepts, while for park bench, grass and path are primary indicators. This implies that 1) VLMs may exhibit insensitivity to the presence of core attributes; 2) directly adapting to downstream tasks may heavily rely on spurious features for predictions. In fact, this also aligns with findings from concurrent work (Wang et al., 2024), which indicates that VLMs are more susceptible to spurious features compared to unimodal architectures.

In addition to the above observations, this prompts us to consider several questions.

**Where do spurious attributes come from?** In the case of LLMs, this phenomenon may be attributed to statistical bias in their large-scale training data. In practical scenarios, when describing a complex object, there may be a tendency to focus more on its accompanying scenes and associated elements rather than its core components. However, these accompanying elements may not contribute to a VLM's generalizable understanding of a specific category. Conversely, regarding vocabulary-based methods, their attribute selection heavily relies on in-distribution samples, and this preference for attributes may be also detrimental to generalization.

**Is there a better way to identify spurious attributes?** While manually purifying the attribute pool may enhance generalization, it faces two primary challenges: 1) scalability issues as the dataset size grows, and 2) it is a simple solution lacking quantitative assessment of the spurious correlation of each attribute, potentially leading to false positives, *i.e.*, attributes that co-occur with the category merely by chance. Hence, we also experiment with an LLM-assisted variant called ArGue*, which adjusts the LLM prompts to encourage a stronger focus on the category itself. However, as demonstrated empirically in Table 1, the reduction in the spurious rate is modest (6.06% → 5.76%), resulting in only marginal gains (66.07% → 66.33%).

### 3.3 SPURIOUS ATTRIBUTE PROBING

Motivated by the above considerations, we introduce SPURIOUS ATTRIBUTE PROBING (SAP), an approach to creating a comprehensive attribute pool where spurious and core attributes are distinctly separated. Initially, SAP utilizes Multi-modal Large Language Models (MLLMs) to differentiate attributes belonging to the target category, distinguishing core attributes from non-core counterparts. To determine if the coexistence of the latter with the category is coincidental or correlated, Concept Bottleneck Models (CBMs) gauge their impact on VLMs' decision-making, with high-influence ones being identified as spurious attributes. By leveraging SAP, a pure and robust attribute pool is achieved, significantly improving the generalization of existing attribute-based methods.

**Prompting MLLMs.** Here we assume $\mathcal{U}$ as a MLLM, differing from prior methods that only accept textual prompts. $\mathcal{U}$ concurrently processes both prompts and images as input such that $\mathcal{A}_c = \mathcal{U}(\mathcal{X}_c, \mathcal{H}(c))$, where $\mathcal{X}_c$ represents training images labeled with $c$, and $\mathcal{H}(c)$ is a set of chain-of-thought prompts probing two aspects: core attributes and non-core counterparts. Specifically, we design three question formats:

> **Q1:** List all the visual cues you see in the photo:
> **Q2:** Are the objects you list a part of ___?
> **Q3:** Describe ___ in the photo in details:

Combining Q1 and Q2 helps identify non-core attributes in the images, while Q3 provides detailed core attributes belonging to the category. Empirically, we'll use multiple templates for each question to ensure thoroughness. Upon reformulation, we derive $\mathcal{A}_c = \widetilde{\mathcal{A}}_c^- \cup \mathcal{A}_c^+$, where $\widetilde{\mathcal{A}}_c^-$ denotes non-core attributes, and $\mathcal{A}_c^+$ represents the core ones.

**Finding spurious attributes.** Given non-core attributes $\widetilde{\mathcal{A}}_c^-$, we use their weights on model predictions as a proxy to indicate the extent of spurious correlations. We use a CBM (Koh et al., 2020) to achieve this goal. Specifically, we construct a bottleneck embedding $\mathcal{E} \in \mathbb{R}^{N \times d}$ against the attribute pool $\mathcal{P}$, where $N = |\mathcal{C}| \times J$ indicates the total number of attributes in the current pool, each row $\mathcal{E}_i \in \mathbb{R}^d$ indicates the feature of the corresponding attribute, and $d$ is the feature dimension. In other words, $\mathcal{E}$ is a feature matrix that concatenates attributes across all the categories together. The procedure of CBMs is to combine two functions to make the prediction: $\hat{c} = h(g(\phi_I(x), \mathcal{E}))$, where $g: \mathbb{R}^d \times \mathbb{R}^{N \times d} \to \mathbb{R}^N$ measures the score between the image feature and every attribute in the bottleneck, $h: \mathbb{R}^N \to \mathcal{C}$ produces the final prediction based on the score vector. Following Yang et al. (2023), we set the score vector $g$ as the dot product of the features between two modalities $g(\phi_I(x), \mathcal{E}) = \phi_I(x) \cdot \mathcal{E}^\intercal$, and $h$ as the linear projection with a learnable weight matrix $\mathcal{W} \in \mathbb{R}^{|\mathcal{C}| \times N}$ such that $h(g; \mathcal{W}) = softmax(g \cdot \mathcal{W}^\intercal)$. Intuitively, $\mathcal{W}_{ij}$ indicates the impact factor of $j^{th}$ attribute, *i.e.*, $a_i^j$ on the prediction $i$. To learn the weights of attributes on predictions, a cross-entropy loss is typically employed.

For non-core attributes, a high weight indicates a strong correlation with the category, whereas a low weight suggests that its presence might be coincidental. Thus, a natural idea to confirm spurious attributes is by thresholding $\gamma$. Formally, for a specific prediction $i$, the spurious attributes are defined as

$$\mathcal{A}_i^- = \{a_i^j \in \widetilde{\mathcal{A}}_i^- \mid \mathcal{W}_{ij} \geq \gamma\}. \tag{2}$$

There is a trade-off in choosing $\gamma$. If $\gamma$ is too large, some spurious attributes may be overlooked. Conversely, if it is too small, a large number of false positives may be introduced. Additionally, we observe significant variability in attribute weight distributions among different categories, posing challenges in identifying spurious attributes with a uniform threshold. Creating a manual threshold for each category is prohibitively expensive. Hence, we introduce an adaptive strategy. Given a prediction $c$, we select $\gamma_c$ as the lowest weight of $\mathcal{A}_c^+$ such that non-core attributes with weights higher than any of the core attributes are considered spurious. This ensures flexible selection of spurious attributes, greatly aiding SAS introduced in Section 3.4 to be discussed next.

### 3.4 SPURIOUS ATTRIBUTE SHIELDING

SAP complements existing attribute-based methods by screening out spurious attributes from the pool, while it does not prevent the PEFT family from learning spurious features in the training im-

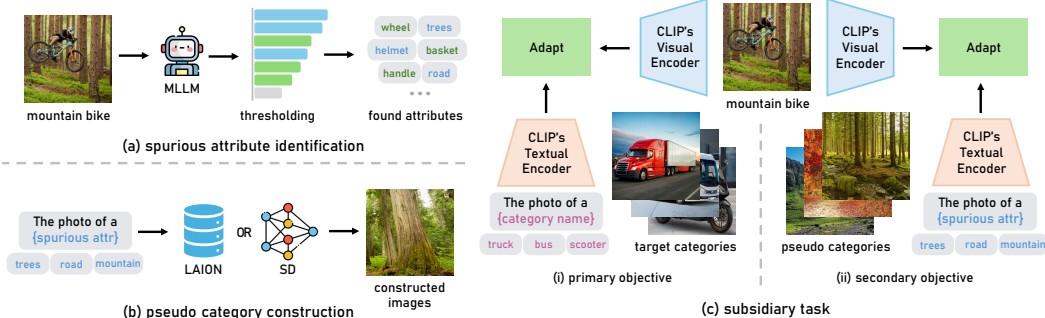

Figure 2: **The overview of** `SAS`. In **(a)**, we generate and identify spurious attributes with `SAP`. In **(b)**, we construct pseudo categories by synthetic data (SD) or retrieval (LAION). In **(c)**, apart from the main objective (i), *e.g.*, cross-entropy loss, we introduce an auxiliary subsidiary task (ii) for learning robust features.

ages. Hence, we propose SPURIOUS ATTRIBUTE SHIELDING (`SAS`), a plug-and-play module to be seamlessly integrated into arbitrary PEFT methods by mitigating the influence of spurious features. Building upon the spurious attributes detected by `SAP`, `SAS` introduces a subsidiary task by constructing a set of pseudo categories alongside the real one and let VLMs differentiate among them. This auxiliary learning objective effectively prompts VLMs to learn robust features rather than ones referred by these spurious attributes. For instance, if streetlight is a spurious attribute for the category vehicle impacting decision-making significantly, we introduce a pseudo category specifically for streetlight and differentiate between the two, thereby reducing the reliance of streetlight when identifying vehicle. Fig. 2 demonstrates the overall pipeline of `SAS`.

Formally, given a category $c$, we establish a set of pseudo categories $\mathcal{J}_c$ with constructed images $\{\hat{\mathcal{X}}_j \mid j \in \mathcal{J}_c\}$. Thus we define a subsidiary dataset $\mathcal{D}_c = \{(x, y) \mid x \in \hat{\mathcal{X}} \cup \mathcal{X}_c \text{ and } y \in \mathcal{J}_c \cup \{c\}\}$. We aim to optimize the following

$$\mathcal{L}_{pse} = -\sum_{c \in \mathcal{C}} \mathbb{E}_{(x,y) \in \mathcal{D}_c} \log \frac{\exp(s(\phi_I(x), \phi_L(t_y))/\tau)}{\sum\limits_{j \in \mathcal{J}_c \cup \{c\}} \exp(s(\phi_I(x), \phi_L(t_j))/\tau)}. \tag{3}$$

That is, we introduce an additional cross-entropy loss for classifying between each target category and its corresponding pseudo categories, which are defined by spurious attributes. This can be viewed as a subsidiary task, aimed at reducing reliance on spurious attributes while achieving correct classification in the downstream task. When integrated with existing methods, we introduce a scaler $\lambda$ to balance the importance of $\mathcal{L}_{pse}$: $\mathcal{L}_{tot} = \mathcal{L}_{ce} + \lambda \mathcal{L}_{pse}$.

A natural question to ask is: how to construct $\hat{\mathcal{X}}$ such that the adapted modules may effectively distinguish spurious attributes from the target categories? In this work, we introduce two approaches.

**Synthetic Generation.** We create pseudo categories using synthetic data by leveraging the text-to-image model Stable Diffusion (SD) (Rombach et al., 2022). We consider two key factors: 1) diversity: Our goal is for pseudo categories to fully represent the features of spurious attributes. To achieve this, we use LLMs to generate various prompts, which are then used as inputs to SD to produce a range of images. 2) purity: If the constructed images contain not only spurious attributes but also unexpected elements, *i.e.*, noise attributes, these noise attributes may create new shortcuts, affecting the effectiveness of `SAS`. Empirically, selecting the top-k images that are most similar to the corresponding spurious attribute can help reduce the presence of noise attributes. Further details are in Supp. Mat. A.

**Pretraining Retrieval.** An alternative way is to gather image samples from pre-training data such as LAION-5B (Schuhmann et al., 2022), a publicly available subset of CLIP's pre-training datasets. We use captions as a proxy to efficiently determine semantic similarity between pre-training images and spurious attributes. Finally, we select the top-k matches to the spurious attributes to create the pseudo categories.

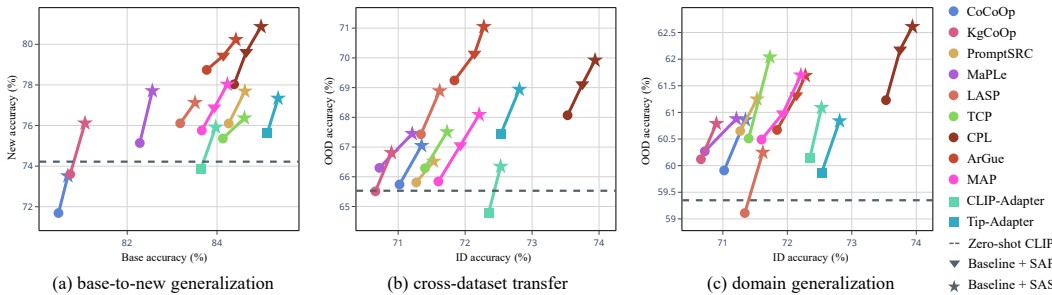

Figure 3: **The average results of three generalization tasks over 11 datasets.** The x-axis and y-axis represent in-distribution/base accuracy and out-of-distribution/new accuracy, respectively. We present the out-of-distribution accuracy of vanilla CLIP as a horizontal bar to represent the zero-shot capability. The detailed numerical results are provided in Supp. Mat. E.

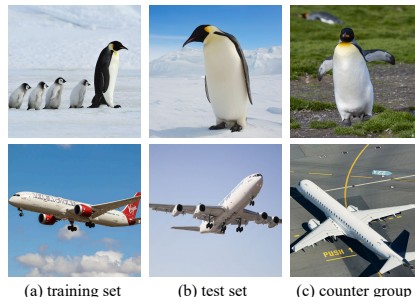

(a) training set        (b) test set        (c) counter group

Figure 4: **Example samples from test set and counter group.** The samples from counter group do not contain spurious attributes, *e.g.*, ice or sky.

| Method | ImageNet | | OxfordPets | | FGVCAircraft | |
|---|---|---|---|---|---|---|
| | Test | Counter | Test | Counter | Test | Counter |
| PSRC | 72.46 | 63.06 | 87.51 | 74.08 | **39.72** | 21.35 |
| PSRC + SAS | **73.82** | **68.55** | **88.27** | **77.17** | 39.54 | **26.43** |
| TCP | 71.54 | 60.17 | 89.11 | 72.84 | 38.30 | 23.61 |
| TCP + SAS | **72.81** | **64.50** | **89.86** | **76.14** | **39.23** | **28.44** |
| CPL | 69.08 | 64.75 | **90.32** | 79.91 | 40.43 | 27.65 |
| CPL + SAS | **70.15** | **67.98** | 90.12 | **83.45** | **40.64** | **32.12** |

Table 2: **The results for standard few-shot classification on test set and counter group, respectively.** Essentially, counter group is a subset of test set where spurious attributes are removed.

## 4 EXPERIMENT

**Task Setting.** Following previous work (Khattak et al., 2023a; Zhou et al., 2022b; Khattak et al., 2023b), the experiment is conducted on base-to-new generalization, cross-dataset transfer and domain generalization. For base-to-new generalization, the datasets are equally divided into base and new categories, where the model is trained on base categories and evaluated on unseen ones. For cross-dataset transfer, the model will be trained on a large-scale dataset, and generalized across various other datasets. For domain generalization, the model will be transferred from an in-distribution dataset to several variants.

**Datasets.** For base-to-new generalization, we employ 11 datasets, including ImageNet (Deng et al., 2009), Caltech101 (Fei-Fei et al., 2004), OxfordPets (Parkhi et al., 2012), StanfordCars (Krause et al., 2013), Flowers102 (Nilsback & Zisserman, 2008), Food101 (Bossard et al., 2014), FGVCAircraft (Maji et al., 2013), SUN397 (Xiao et al., 2010), UCF101 (Soomro et al., 2012), DTD (Cimpoi et al., 2014) and EuroSAT (Helber et al., 2019). For cross-dataset transfer, we train models on ImageNet (Deng et al., 2009), and evaluate on the remaining datasets mentioned above. For domain generalization, we designate ImageNet as the in-distribution dataset, with four out-of-distribution variants encompassing ImageNetV2 (Recht et al., 2019), ImageNet-Sketch (Wang et al., 2019), ImageNet-A (Hendrycks et al., 2021b) and ImageNet-R (Hendrycks et al., 2021a). The experiments are carried out in the few-shot setting, where we randomly sample 16 shots for each category to compose the training set.

**Baselines.** We consider various PEFT approaches. Specifically, for prompt tuning, we consider category conditioning including CoCoOp (Zhou et al., 2022b) and TCP (Yao et al., 2024), regularization techniques encompassing KgCoOp (Yao et al., 2023), LASP (Bulat & Tzimiropoulos, 2023) and PromptSRC (Khattak et al., 2023b), attribute-based methods such as CPL (Zhang et al., 2024c), ArGue (Tian et al., 2024) and MAP (Liu et al., 2024b). We also consider multi-modal prompt tuning, *i.e.*, MaPLe (Khattak et al., 2023a). Besides, CLIP-Adapter (Gao et al., 2024) and its training-free

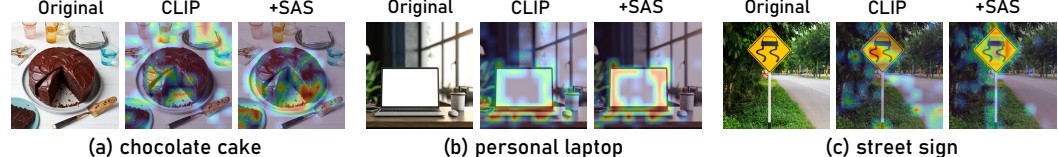

Figure 5: **The saliency map of VLMs with and without SAS.** From left to right we show three example categories, including chocolate cake, personal laptop, and street sign.

| #p | 1 | 2 | 3 | 4 | 5 | 6 | 7 |
|---|---|---|---|---|---|---|---|
| Base | 83.10 | 83.47 | 83.41 | 83.59 | 83.64 | 83.51 | **83.68** |
| New | 75.05 | 75.89 | 76.91 | 77.19 | 77.36 | **77.41** | 77.23 |
| HM | 78.87 | 79.50 | 80.03 | 80.26 | **80.38** | 80.34 | 80.32 |

| $\gamma$ | 0.0 | 0.2 | 0.4 | 0.6 | 0.8 | 1.0 | Ada. |
|---|---|---|---|---|---|---|---|
| Base | 82.95 | 83.16 | 83.53 | 83.43 | 83.28 | 83.23 | **83.64** |
| New | 76.21 | 76.60 | 76.41 | 76.01 | 75.30 | 74.82 | **77.36** |
| HM | 79.44 | 79.75 | 79.81 | 79.55 | 79.09 | 78.80 | **80.38** |

Table 3: **Varying number of SD prompts on base-to-new generalization.** The results are averaged across 11 datasets and 11 baselines.

Table 4: **Varying $\gamma$ on base-to-new generalization.** We experiment with fixed values, juxtaposing them against the suggested adaptive strategy.

| Method | Epoch | Time ↓ | Accuracy ↑ | Gain ↑ |
|---|---|---|---|---|
| ZSCLIP | — | — | 70.22 | — |
| CoCoOp | 10 | 4h37m | 73.10 | — |
| + SAS | 10 | 6h18m | 74.21 | +1.11 |
| + selective trick | 10 | 4h51m | 74.02 | +0.92 |
| PromptSRC | 50 | 3h29m | 74.01 | — |
| + SAS | 50 | 4h56m | 75.46 | +1.45 |
| + selective trick | 50 | 3h38m | 75.20 | +1.19 |

Table 5: **The efficiency of SAS with and without selective trick.** We evaluate in terms of training time and accuracy gains given the same number of epochs. We opt for two time-intensive baselines, *i.e.*, CoCoOp and Prompt-SRC, and train both on ImageNet under base-to-new generalization task.

version Tip-Adapter (Zhang et al., 2022b) are involved. All results are averaged over three runs with distinct initialization.

**Implementation Details.** Unless specified otherwise, we use pre-trained CLIP (Radford et al., 2021) and ViT-B16 as the visual backbone for fair comparison. Since our proposed method is a plug-and-play module, we strictly adhere to the settings of existing works, including optimizers, batch size, learning rate, and other strategies. This indicates that for different baselines, we may use distinct hyperparameters specified in their respective papers. For SAP, we use GPT-4V Turbo (Achiam et al., 2023) as the MLLM, with a temperature scaler of 0.7 and an image understanding level set to *high*. For SAS, by default, we use ChatGPT to generate 5 prompts for each spurious attribute, which are then fed into Stable Diffusion (Rombach et al., 2022) to create pseudo categories. More details, such as the effect of choices of MLLMs and comparison between pseudo category construction with synthesized and pre-training data, are provided in Supp. Mat. B. Each pseudo category contains 16 shots, matching the number in the target category. All experiments are conducted on a single NVIDIA 4090 GPU.

## 4.1 Main Results

**Our method is complementary to PEFT approaches.** Fig. 3 depicts the results of baselines and their integration with SAP and SAS. We observe an upward trend in accuracy, indicating improvements in out-of-distribution accuracy without compromising downstream task performance. For conventional methods, *e.g.*, CoCoOp, SAS helps achieve zero-shot capability on distribution shifts. For strong baselines, *e.g.*, CPL, the incorporation of SAP and SAS enables them to reach a new state-of-the-art benchmark. Overall, applying our method leads to an average improvement of over 2% in most baselines. These promising results align with the observation of the biased nature of VLMs, as demonstrated in Section 3.2.

**Our method is effective on counter test samples.** To further highlight the effectiveness of SAS, we conduct standard few-shot classification in an adversarial evaluation manner. This involves selecting a subset from the original test set to create a counter group for evaluating the VLM. For each category, we filter out images from the test set that bear high semantic similarity to the identified spurious attributes, retaining only images free of such attributes. This counter group is significantly

more challenging to predict using spurious attributes compared to the entire test set. Fig. 4 displays example images from the test set and counter group. Table 2 presents the improvement in accuracy of SAS over baselines, both for the test set and counter group. We notice that 1) for the counter group, its accuracy is much lower than that of the test set; 2) SAS effectively bridges this gap, with improvement on the counter group far exceeding that on the test set, up to approximately 6%.

## 4.2 ABLATION STUDY

**Diverse construction is beneficial for learning robust features.** During training, we aim for the constructed pseudo-categories to possess similar semantics to their target counterparts, thereby creating a strong contrast, while also maintaining high diversity to comprehensively represent spurious features. This trade-off is achieved by varying the number of SD prompts. As shown in Table 3, the effectiveness of SAS in assisting the baselines becomes more evident with an increasing number of prompts. This underscores the importance of the quality of pseudo-categories, which should thoroughly reflect the corresponding spurious attributes.

**Selecting appropriate spurious attributes matters.** The core principle of SAS is to introduce auxiliary categories to be trained alongside the main task, preventing the model from achieving high accuracy through spurious features. A natural concern is whether the model's gains are due to the introduction of additional data rather than an increase in robustness. In other words, does the model genuinely learn to distinguish spurious attributes from pseudo categories? We investigate this by adjusting the threshold $\gamma$ to control the presence of spurious attributes in pseudo categories. A higher $\gamma$ indicates a shortage of identified spurious attributes, while a lower $\gamma$ may introduce false positives. In Table 4, we observe that performance significantly drops when $\gamma$ is either too high or too low. This indicates that 1) spurious attributes play a crucial role in the contribution of SAS, and 2) the introduction of noisy attributes actually impairs the model's robustness. Additionally, the suggested adaptive strategy, which allows for flexible selection of spurious attributes, outperforms the pre-defined $\gamma$.

## 4.3 FURTHER ANALYSIS

**SAS corrects the preference of VLMs on spurious attributes.** To qualitatively assess SAS's impact on VLMs, we present the saliency maps of VLMs with and without SAS in Fig. 5. Common spurious correlations can be observed, such as (a) utensils appearing alongside chocolate cake, and (b) a mouse typically appearing with a laptop. In critical applications, such as autonomous driving, (c) road tends to act as confounders for street sign. SAS can effectively shift attention from these spurious attributes to the corresponding main objects. While we revisit Fig. 1(c), this also aligns with the interpretation of CBMs that SAS suppresses the influence of spurious attributes on predictions.

**Balancing the trade-off between efficiency and effectiveness.** One potential concern with SAS is its impact on training efficiency. Applying a distinct loss to each category can be computationally demanding. To address this, we introduce a selective optimization trick. Rather than targeting all categories, we only optimize ones that heavily rely on spurious attributes for predictions. Details of this approach are outlined in Supp. Mat. C. In Table 5, we demonstrate the effectiveness of this selective strategy by optimizing only 10% of the categories, showing the training time and accuracy. This approach significantly reduces SAS's training time while preserving most of its accuracy gains.

## 5 CONCLUSION

This paper is motivated by an often-overlooked fact: VLMs tend to favor spurious attributes in their predictions, leading to decreased accuracy on out-of-distribution datasets. To tackle this issue, we first introduce SPURIOUS ATTRIBUTE PROBING (SAP), which identifies and filters out these problematic attributes, significantly improving the generalization of existing attribute-based methods. Furthermore, to alleviate the biased nature of VLMs, we introduce SPURIOUS ATTRIBUTE SHIELDING (SAS), a plug-and-play module that reduces the influence of these attributes on predictions and complements various PEFT approaches. Both solutions significantly enhance accuracy in handling distribution shifts without compromising performance on downstream tasks, achieving a new state-of-the-art level.

ACKNOWLEDGEMENT

This research was, in part, funded by the U.S. Government – DARPA TIAMAT HR00112490421. The views and conclusions contained in this document are those of the authors and should not be interpreted as representing the official policies, either expressed or implied, of the U.S. Government.

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

CONTENTS

# A  IMPLEMENTATION DETAILS

## A.1  FINDING SPURIOUS ATTRIBUTES

We delve into our manual identification process for spurious attributes as described in Section 3.2. Following the approach outlined in (Singla & Feizi, 2021), we present a simplified version. For each category, we randomly select 5 images from the training set and generate the corresponding heatmap. We also reference external sources like Wikipedia and seek advice from ChatGPT. Using this information, we assess whether an attribute belongs to the main object or a separate background element, with options: "Yes", "No", or "Unsure". Finally, attributes categorized as "No" are deemed spurious attributes. It is important to mention that unlike (Singla & Feizi, 2021), we do not conduct crowd studies. All supervision tasks are performed by the authors.

## A.2  PROMPTING LLMS

We conduct a naive attempt to modify the prompting technique of LLMs to avoid generating spurious attributes in Section 3.2. We try three variant prompt templates by appending or inserting additional instructions as follows:

**T1:** Only focus on ___ itself.
**T2:** Imagine you are an expert of ___.
**T3:** Do not describe other than ___.

For each instruction, we position it at either the beginning or the end, yielding six combinations. Then, we employ existing attribute-based methods, *e.g.*, ArGue (Tian et al., 2024), to derive results, averaging them across all combinations.

## A.3  QUERYING MLLMS

In addition to the techniques and parameters introduced in the main paper, we believe a crucial step in dealing with MLLMs is managing their outputs. Given a specified temperature, the output variance of an MLLM, particularly GPT-4V, for the same input can be significant. The responses may range from a single word to a complete paragraph, and the model may fail to follow the demonstrated formats or refuse to respond. Similar challenges have been noted in recent studies, such as DCLIP (Menon & Vondrick, 2023) and CuPL (Pratt et al., 2023), when using MLLMs or LLMs to generate attributes. In this work, we employ a simple regular expression to retain responses of suitable length and exclude those that are not formatted with bullet points. Additionally, we filter out duplicate or similar attributes. For example, between ice surface and glacier we typically randomly select only one.

## A.4  CONSTRUCTING PSEUDO CATEGORIES

Here, we describe the process of constructing images targeting spurious attributes using SD (Rombach et al., 2022) or LAION-5B (Schuhmann et al., 2022). For the former, following Sus-X (Udandarao et al., 2022), we use the common checkpoint stable-diffusion-v1-4, with a guidance scale of 7.0. The diffusion step is set to 100, with a fixed output resolution of 512x512. Additionally, to ensure the diversity of the images, we use ChatGPT to generate multiple SD prompts. Specifically, we provide a vanilla prompt as an example, e.g., a photo of a mouse, and then ask GPT to rephrase the prompt in different formats. Some example generated prompts are displayed below.

**P1:** a 3D realistic photo of a ___
**P2:** a high-quality natural image of ___.
**P3:** a intriguing portray of ___.

It is worth noting that the prompts mentioned above are also applicable for pre-training retrieval. For LAION-5B, we select the matches with the highest average semantic similarity to these GPT-generated prompts to construct pseudo categories. This approach ensures the diversity of the retrieved images while also enhancing the reliability of semantic matching.

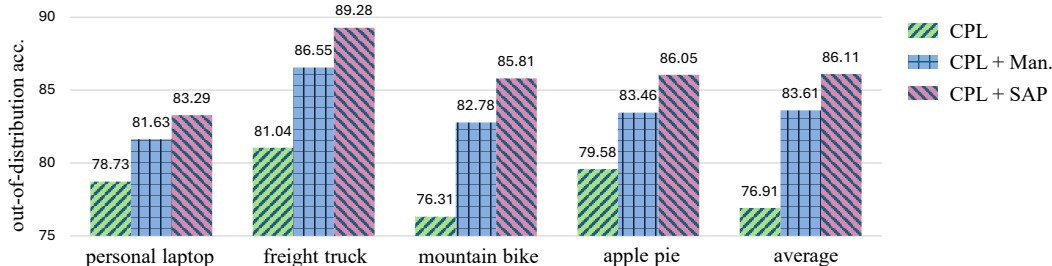

Figure 6: **The per-category out-of-distribution accuracy on domain generalization.** In this setting, based on the strong baseline CPL, we remove spurious attributes identified by manual inspection (Man.) or SAP for a specific category and compare the accuracy change on the category in the out-of-distribution datasets. All results are averaged over 4 ImageNet variants.

| Category name | Spurious attributes | Average weights |
|---|---|---|
| Personal laptop | *mouse, coffee, charger, worktable* | 77.34% / 46.73% |
| Freight truck | *road, traffic light, trees, street* | 82.16% / 54.69% |
| Mountain bike | *trees, road, mountain, swamp* | 74.81% / 43.02% |
| Apple pie | *fork, plates, dining car, tablecloth* | 67.29% / 37.44% |

Table 6: **The spurious attributes identified by SAP.** For each example category, we pinpoint its spurious attributes and determine the average attribute weights on model predictions using CBMs across identified spurious attributes (**Left**) and all generated attributes (**Right**).

## B  MORE EVALUATION

### B.1  EXAMPLE SPURIOUS ATTRIBUTES

SAP quantifies the identification of spurious attributes without human supervision, offering a more precise measure of spurious correlation. This aids in effectively pinpointing attributes favored by VLMs. Table 6 showcases typical spurious attributes found by SAP, including instances like mouse frequently appearing with laptop, or fork being closely associated with apple pie. Additionally, we assess their weights on model predictions, along with the average weights of all generated attributes for reference. Notably, spurious attributes carry substantially higher weights in model decision-making compared to overall attributes, further underscoring the biased nature of VLMs.

### B.2  SAP VS HUMAN SUPERVISION

Finding spurious attributes through human supervision (Singla & Feizi, 2021; Wong et al., 2021), while comprehensive, has significant drawbacks: 1) it incurs extremely high labor costs; 2) its strong subjectivity easily introduces false positives, where identified attributes are only present by chance. Here, we compare the performance of the proposed automatic identification method, SAP, with human supervision. We adopt domain generalization as the task and select CPL (Zhang et al., 2024c) as the baseline. For better interpretation, we remove spurious attributes from individual categories one at a time and record the change in per-category accuracy on out-of-distribution datasets. Fig. 6 depicts the results in CPL, as well as the results after removing spurious attributes through the two identification approaches. It can be seen that SAP's performance is comparable or even outperforms human supervision.

### B.3  QUERYING WITH SAP AT SCALE

In the main paper, we address a challenging setting, specifically few-shot scenarios where training data is limited. This leads to a pertinent question: is a small number of images truly adequate for SAP to identify spurious attributes within categories? In other words, would querying more images further enhance SAP's performance? To investigate the potential of scaling up, we expand

| # Query Images | 1 | 2 | 4 | 8 | 16 | 32 | 64 |
|---|---|---|---|---|---|---|---|
| CoCoOp | 75.62 | 76.06 | 76.84 | 77.56 | 78.10 | 78.24 | 78.28 |
| MaPLe | 81.98 | 82.55 | 83.47 | 84.01 | 84.49 | 84.60 | 84.53 |
| PromptSRC | 82.59 | 83.35 | 84.13 | 84.72 | 85.46 | 85.63 | 85.68 |

Table 7: **The evaluation on base-to-new generalization while querying different number of images per-category.** The results are averaged across 11 datasets.

| MLLM | BLIP-2 | LLaVA | InternVL | GPT-4V |
|---|---|---|---|---|
| CoCoOp | 72.28 | 72.79 | 73.14 | **73.50** |
| MaPLe | 76.20 | 76.83 | 77.25 | **77.69** |
| PromptSRC | 76.43 | 77.01 | 77.37 | **77.88** |

Table 8: **The evaluation on base-to-new generalization with various MLLMs.**

the training dataset from 16-shot to 256-shot and have GPT-4V query 1, 2, 4, 8, 16, 32, and 64 randomly selected images from the training shots. We evaluate the average new category accuracy on base-to-new generalization tasks across 11 datasets, comparing three typical baselines: CoCoOp, MaPLe, and PromptSRC. As shown in Table 7, despite the availability of additional shots during training, the results tend to plateau when querying with 16 images. This indicates that even with an expanded training dataset, MLLMs require only around 16 query images to capture sufficient and effective spurious attributes.

### B.4 Effect of Choices of MLLMs

In previous experiments, we default our MLLM to GPT-4V. Here, we attempt to use more open-sourced MLLMs to comprehensively evaluate the robustness of our proposed method. We consider three recently popular MLLMs: BLIP-2 (Li et al., 2023a), LLaVA (Liu et al., 2024a), and InternVL (Chen et al., 2024). Table 8 presents the performance of these different MLLMs on base-to-new generalization tasks with 16-shot. As expected, GPT-4V, the proprietary model, achieves the best results. The next best performance is from InternVL. Conversely, BLIP-2 shows the poorest performance, which we attribute to its tendency to produce a limited vocabulary that results in overly broad core and spurious attributes.

### B.5 Synthetic Generation vs Pre-training Retrieval

In previous experiments, our default approach is to utilize Stable Diffusion for constructing pseudo categories. Here, we explore an alternative method: retrieving image samples from the pre-training dataset. Table 9 illustrates the results of both approaches across several baselines. Notably, Stable Diffusion consistently outperforms retrieval from LAION-5B. This unexpected result is intriguing,

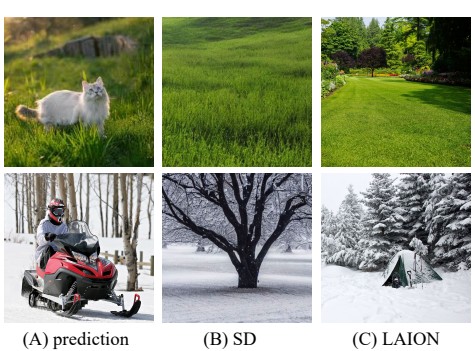

(A) prediction    (B) SD    (C) LAION

Figure 7: **Constructed images.**

|       | CoCoOp | KgCoOp | MaPLe | PromptSRC | CPL   |
|-------|--------|--------|-------|-----------|-------|
| SD    | **76.93** | **78.51** | **80.06** | 80.57     | **82.87** |
| LAION | 76.46  | 78.45  | 79.32 | **80.85**   | 82.35 |

Table 9: **The comparison between synthetic generation and pre-training dataset retrieval.** We select 5 strong baselines. The results are harmonic mean of accuracy on base and new categories for base-to-new generalization.

| $\lambda$ | 0 | 1 | 2 | 5 | 10 | 20 |
|-----------|------|------|------|------|------|------|
| Base | 83.23 | **83.73** | 83.64 | 83.15 | 82.41 | 81.53 |
| New  | 74.82 | 76.00 | 77.36 | **77.73** | 76.60 | 76.89 |
| HM   | 78.80 | 79.68 | **80.38** | 80.35 | 79.40 | 79.14 |

Table 10: **The effect of $\mathcal{L}_{pse}$ with different $\lambda$ on base-to-new generalization.**

considering that pre-training images predominantly consist of real data, which one would expect to better match the style of the target category. However, the results suggest otherwise. We speculate that the complexity of real image distributions, coupled with noise attributes, may contribute to this disparity. For instance, in Fig. 7, when associating the spurious attribute snowforest with snowmobile, the top-1 match retrieved using LAION-5B includes elements such as tent and bag. These noise attributes could potentially introduce new shortcuts, complicating the model's ability to differentiate spurious attributes from the target category.

## B.6 BALANCING THE EFFECT OF SAS

Table 10 examines the balancing effect between $\mathcal{L}_{pse}$ and primary learning objectives in existing work in terms of $\lambda$. The best trade-off is observed at around $\lambda = 2$. As $\lambda$ increases further, it begins to neglect the primary objectives of the baselines, leading to a decline in base accuracy. Notably, these results are averaged across multiple baselines. In fact, for distinct baselines, we suggest exploring optimal values individually due to their respective learning characteristics.

## B.7 QUANTITATIVE COMPARISON WITH RELATED WORK

In the main text, we primarily demonstrate the effectiveness of SAS in complementing existing PEFT methods. Here, we further substantiate the advantages of SAS by comparing it with other state-of-the-art spurious correlation mitigation approaches. We evaluate a typical property of VLMs, group robustness, which indicates the invariance of VLMs under different associations between labels and attributes. For the baselines, we consider C-Adapter (Zhang & Ré, 2022) and CFR (You et al., 2024),

| Method | Waterbirds | | | CelebA | | | BREEDS | | | CIFAR-10.02 | | |
|--------|------|------|------|------|------|------|------|------|------|------|------|------|
| Accuracy (%) | WG | Avg | Gap | WG | Avg | Gap | WG | Avg | Gap | WG | Avg | Gap |
| Zero-shot | 25.7 | 87.3 | 61.6 | 62.1 | 71.9 | 9.8 | 4.0 | 86.6 | 82.6 | 72.0 | 93.2 | 21.2 |
| RoboShot | 45.2 | 79.9 | 34.7 | 82.6 | 85.5 | 2.9 | 56.4 | 80.3 | 23.9 | 79.1 | 95.6 | 16.5 |
| Linear Probe | 65.9 | 97.6 | 31.7 | 28.3 | 94.7 | 66.4 | 84.0 | 98.6 | 14.6 | 87.5 | 96.1 | 8.6 |
| C-Adapter | 86.9 | 96.2 | 9.3 | 84.6 | 90.4 | 5.8 | 80.0 | 97.5 | 17.5 | 82.2 | 96.1 | 13.9 |
| DISC | 88.7 | 93.8 | 5.1 | 82.0 | 92.5 | 10.5 | 86.3 | 95.8 | 9.5 | 84.7 | 94.3 | 9.6 |
| CFR | 88.2 | 96.7 | 8.5 | 84.7 | 87.8 | 3.1 | 85.0 | 96.1 | 11.1 | **89.1** | 92.5 | 3.4 |
| SAS | **89.7** | 96.3 | 6.6 | **87.4** | 91.1 | 3.7 | **87.8** | 96.4 | 8.6 | 88.5 | 95.2 | 6.7 |

Table 11: **The group robustness evaluation of SAS and other spurious correlation mitigation methods.** We report worst-group accuracy (WG), average-group accuracy (Avg) and the gap between. Note that RoboShot is a zero-shot calibration method, while other approaches are training-required.

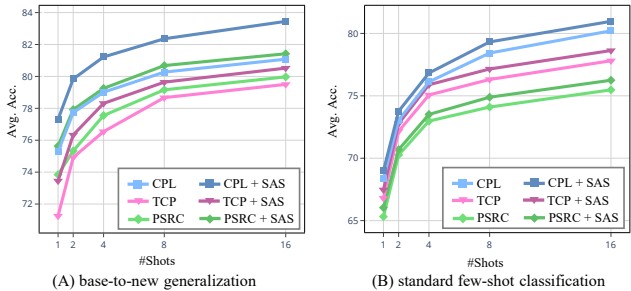

Figure 8: **The results varying shots on base-to-new generalization and few-shot classifcation.**

where spurious attributes are assumed to be unknown. We also include RoboShot (Adila et al., 2023) and DISC (Wu et al., 2023), where, similar to our approach, spurious concepts are identified and used for precise mitigation. By default, we configure SAS to optimize only the learnable textual prompt, *i.e.*, CoOp (Zhou et al., 2022a). It is worth noting that RoboShot (Adila et al., 2023) is a zero-shot approach that calibrates pre-trained embeddings. Following Zhang & Ré (2022), we consider four datasets with group annotations: Waterbirds (Sagawa et al., 2019), CelebA (Liu et al., 2018), BREEDS Living-17 (Santurkar et al., 2020), and CIFAR-10.02 (Krizhevsky et al., 2009). In Table 11, average-group accuracy, worst-group accuracy, and their gap are reported. It can be observed that SAS achieves a new state-of-the-art in worst-group accuracy across most datasets without excessively compromising average-group accuracy.

## B.8 GENERALIZATION UNDER LIMITED SHOTS

We consider generalization capability in extreme cases, where the shots are further limited, *i.e.*, 1/2/4/8 shots. It is noteworthy that in this scenario, limitations arise from both the insufficient amount of data and the impact on SAP's precision to identify spurious attributes, further affecting SAS performance. We select three strong baselines, encompassing PromptSRC (Khattak et al., 2023b), TCP (Yao et al., 2024) and CPL (Zhang et al., 2024c). Fig. 8 (A) shows the results of combining SAS on base-to-new generalization across different shot settings. It can be seen that the results consistently outperform the original baselines, even only one shot is given.

## B.9 STANDARD FEW-SHOT CLASSIFICATION

We consider the standard scenario where test and training samples originate from the same dataset distribution. Fig. 8 (B) illustrates the results in standard few-shot classification. Notably, integrating SAS does not compromise in-distribution accuracy; instead, it shows a slight and consistent improvement.

## B.10 DISCUSSION OF HYPERPARAMETER SENSITIVITY

Although we observe the state-of-the-art performance of SAS in the main context, an important aspect, hyperparameter sensitivity, still requires discussion. For the newly introduced hyperparameters in SAS, such as $\lambda$ and $\gamma$, their impact on the results has been examined in previous ablation experiments. These experiments reveal that while an optimal value is preferred, SAS is not overly sensitive to these hyperparameters and consistently provides stable improvements within a certain range.

Regarding training hyperparameters such as learning rate and batch size, recent studies (Silva-Rodriguez et al., 2024) have found that some adaptation methods heavily rely on these hyperparameters in few-shot scenarios, complicating practical deployment. In contrast, as shown in Fig. 3 of the main paper, although SAS uses different training hyperparameters for different baselines as specified in the original papers, it consistently achieves gains, demonstrating its robustness to hyperparameters.

| Training Data | CoCoOp | MaPLe | PromptSRC | Average |
|---|---|---|---|---|
| 16-shot main | 70.05 | 75.39 | 73.78 | 72.94 |
| 32-shot main | 71.12 | 76.37 | 75.52 | 74.34 |
| 16-shot main + 16-shot pseudo main | 70.34 | 75.80 | 74.46 | 73.53 |
| 16-shot main + 16-shot pseudo spurious (ours) | **73.50** | **77.69** | **77.88** | **76.36** |

Table 12: **The ablation study on the performance gains**. We introduce two baselines, where the first incorporates additional data from the training set (32-shot main), and the second involves vanilla constructed data from pseudo categories mirroring the main categories (16-shot main + 16-shot pseudo main). In contrast, the pseudo categories of our method feature spurious attributes (16-shot main + 16-shot pseudo spurious)

### B.11 ABLATION ON PERFORMANCE GAINS

In the main paper, we verify the effectiveness of SAS and explore the contribution of spurious attributes to its performance. To further confirm that the performance gains are primarily due to the model's enhanced robustness to spurious attributes rather than additional data, here we conduct a simple ablation study. Specifically, in addition to the proposed method, we design two baselines. In the first baseline, we consider additional data directly from the original dataset featuring the main objects, where we extend the training data from 16 shots to 32 shots (32-shot main). In the second baseline, we involve additional data generated by pseudo categories, where instead of featuring spurious attributes, these pseudo categories are the same as the main categories, i.e., vanilla constructed data (16-shot main + 16-shot pseudo main). In contrast to the first two baselines, our approach creates pseudo categories based on spurious attributes (16-shot main + 16-shot pseudo spurious). For fairness, we ensure that the amount of training data is identical between the two baselines and our approach. We select three typical methods for comparison, including CoCoOp (Zhou et al., 2022b), MaPLe (Khattak et al., 2023a), and PromptSRC (Khattak et al., 2023b), and evaluate them on the base-to-new generalization task. All results are averaged across 11 datasets.

As shown in Table 12, generating additional data using spurious attributes significantly outperforms vanilla constructed data for main categories (76.36% vs 73.53%). Furthermore, our proposed method even exceeds the performance of the 32-shot main (76.36% vs 74.34%). It is important to note that this comparison is not entirely fair for our method, as the latter relies on more labeled data from the original training set. This further suggests that the performance gains are primarily driven by the model's enhanced robustness to spurious attributes, rather than merely the increased training data.

### B.12 MORE VISUALIZATION EXAMPLES

In Fig. 5, we present the saliency maps for some typical categories with and without SAS. For completeness, we provide more examples here. As shown in Fig. 9, SAS consistently reduces VLMs' bias towards spurious cues across various categories, enabling a greater focus on the main objects.

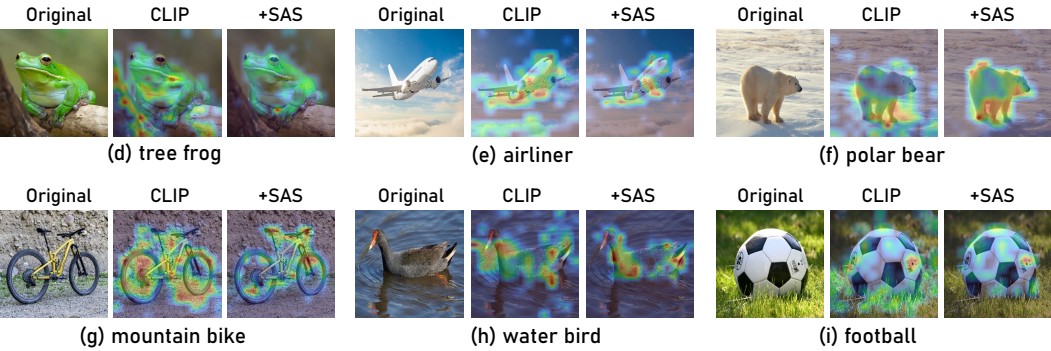

Figure 9: **More saliency map visualization with and without SAS**.

| Model | CoCoOp | MaPLe | PromptSRC | Average |
|---|---|---|---|---|
| BLIP | 68.81 | 72.32 | 72.62 | 70.58 |
| BLIP + SAS | 70.54 | 74.35 | 73.97 | 72.95 |
| CLIPA-v2 | 70.28 | 73.40 | 74.52 | 72.73 |
| CLIPA-v2 + SAS | 72.42 | 74.88 | 77.08 | 74.79 |
| EVA-CLIP | 72.75 | 77.58 | 76.13 | 75.49 |
| EVA-CLIP + SAS | 74.60 | **77.92** | 77.82 | 76.78 |
| SigLIP | 74.99 | 73.78 | 78.64 | 75.80 |
| SigLIP + SAS | **76.41** | 75.26 | **79.87** | **77.18** |

Table 13: **The evaluation results of `SAS` on other VLMs**. We consider four representative VLMs including BLIP, CLIPA-v2, EVA-CLIP and SigLIP.

| Method | Flowers102 | Food101 | FGVCAircraft | StanfordCars | Average Time |
|---|---|---|---|---|---|
| CoCoOp | 10m48s | 18m34s | 8m02s | 12m46s | 12m32s |
| + SAS | 13m14s | 24m07s | 13m36s | 17m08s | 17m01s |
| + selective trick | 11m03s | 20m45s | 10m23s | 14m55s | 14m16s |
| PromptSRC | 6m25s | 15m09s | 5m44s | 9m36s | 9m13s |
| + SAS | 8m26s | 18m21s | 7m11s | 12m04s | 11m30s |
| + selective trick | 6m58s | 16m22s | 6m35s | 10m20s | 10m03s |

Table 14: **The training efficiency of `SAS` and selective optimization on other datasets**.

For example, for the tree frog, SAS reduces VLMs' reliance on tree branches, while for the airliner, the typical spurious attributes are sky or clouds, and the application of SAS alleviates the model's bias towards these elements.

### B.13 MORE VISION-LANGUAGE MODELS

For completeness, here we extend the evaluation of SAS to additional VLMs other than CLIP. Specifically, we select four typical VLMs encompassing BLIP (Li et al., 2022a), CLIPA-v2 (Li et al., 2024), EVA-CLIP (Sun et al., 2023) and SigLIP (Zhai et al., 2023). We record the results on base-to-new generalization, where the setting is consistent with the main paper. As demonstrated in Table 13, our proposed method, SAS, consistently yields performance gains across a range of VLMs, extending beyond just CLIP.

### B.14 COMPUTATIONAL EFFICIENCY AND COST

In this section, we present the computational and time costs of the proposed method, accompanied by a thorough analysis.

**The cost of training**. In the main paper, we present the training time of SAS and the proposed selective optimization trick on ImageNet. Here, furthermore, we provide time statistics for other datasets. The time is measured as runtime of the training script based on the implementation of CoOp (Zhou et al., 2022a). As shown in Table 14, for most datasets, the integration of SAS only increases the training time by approximately 3 to 5 minutes, while selective optimization further reduces this time to a negligible amount. In fact, the selective optimization trick is proposed to address large-scale datasets, such as ImageNet, which contains 1000 categories. For regular datasets ($\sim$ 100 categories), the time consumption of SAS is fully acceptable.

**The cost of diffusion generation**. Here, we provide the estimated inference time required to construct pseudo categories through Stable Diffusion for each dataset. As shown in Table 15, the total inference time is proportional to the size of the dataset, particularly the number of categories involved. For most datasets, the inference time is under half an hour, and the entire inference process

| Dataset | Caltech | Pets | Cars | Flowers | Food | Aircraft | SUN | DTD | EuroSAT | UCF | INet |
|---|---|---|---|---|---|---|---|---|---|---|---|
| Time | 25min | 10min | 45min | 30min | 25min | 30min | 90min | 15min | 5min | 25min | 3h50min |

Table 15: **The diffusion inference time for each dataset**.

| Dataset | Caltech | Pets | Cars | Flowers | Food | Aircraft | SUN | DTD | EuroSAT | UCF | INet |
|---|---|---|---|---|---|---|---|---|---|---|---|
| Time | 10min | 10min | 25min | 10min | 10min | 10min | 35min | 5min | 3min | 10min | 1h30min |

Table 16: **The GPT prompting time for each dataset**.

can be completed within half a day. It is important to note that this is a one-time operation, and no additional inference is needed during subsequent training.

**The cost of GPT prompting**. In our method, a key step is identifying the spurious attributes within each category, which we accomplish by prompting MLLMs, i.e., GPT. Here we provide the time cost of this process along with a thorough analysis. Specifically, to enhance efficiency, we employ batch inference as implemented in Menon & Vondrick (2023), where multiple queries can be processed concurrently, which significantly reduces the inference time for GPT. As shown in Table 16, the GPT inference time for most datasets is under 10 minutes. The complete inference process takes approximately three hours, which is also a one-time operation that does not need to be repeated thereafter. It is worth noting that upon obtaining the responses, we need to perform post-processing such as filtering and selection to determine valid attributes, as detailed in Section A.3, which may require additional time.

### B.15 More Modalities and Tasks

To assess the transferability of our method to other modalities or tasks, we explore video recognition and leave more tasks, such as language reasoning, for future work. Specifically, we choose ViFi-CLIP (Rasheed et al., 2023), a fully fine-tuned CLIP model tailored for video understanding. ViFi-CLIP employs a training framework similar to CLIP, incorporating a temporal pooling layer to derive video representations from multiple frames. Following the base-to-new generalization setting in Rasheed et al. (2023), we evaluate video-level generalization performance on four video datasets: K-400 (Kay et al., 2017), HMDB-51 (Kuehne et al., 2011), UCF-101 (Soomro, 2012), and SSv2 (Goyal et al., 2017). As in the main paper, we select three representative baseline methods: CoCoOp (Zhou et al., 2022b), MaPLe (Khattak et al., 2023a), and PromptSRC (Khattak et al., 2023b). Since ViFi-CLIP shares its architecture with CLIP, these methods can be easily transferred to ViFi-CLIP, which has been implemented by Khattak et al. (2023b). We incorporate the proposed method, SAS, into these baselines to verify its effectiveness by contrasting spurious attributes with each frame of the video. We record the new category accuracy for each dataset which directly reflects the generalization performance on unseen categories. As shown in Table 17, despite the input modalities shifting from images to videos, SAS consistently delivers performance gains across all

| Method | K-400 | HMDB-51 | UCF-101 | SSv2 |
|---|---|---|---|---|
| ViFiCLIP | 61.10 | 53.30 | 67.70 | 12.10 |
| CoCoOp | 64.70 | 54.41 | 68.21 | 14.24 |
| CoCoOp + SAS | 66.39 | 56.64 | 70.40 | 16.01 |
| MaPLe | 64.52 | 58.23 | 70.73 | 14.74 |
| MaPLe + SAS | 66.42 | 59.32 | 72.66 | 16.40 |
| PromptSRC | 68.31 | 62.38 | 76.79 | 17.22 |
| PromptSRC + SAS | **70.23** | **64.70** | **79.31** | **18.95** |

Table 17: **The evaluation results of SAS on four video datasets**. The training is based on ViFi-CLIP, a fully fine-tuned CLIP model for video reasoning.

| Method | ImageNet | Flowers102 | SUN397 | FGVCAircraft | StanfordCars |
|---|---|---|---|---|---|
| MMA | 71.00 | 75.93 | 78.57 | 36.33 | 73.10 |
| MMA + SAS | 72.61 | 77.27 | **80.19** | **37.85** | 75.46 |
| DMN | 72.28 | 78.49 | 77.32 | 32.60 | 74.22 |
| DMN + SAS | **73.34** | **80.17** | 79.74 | 35.38 | **76.30** |

Table 18: **The evaluation of `SAS` on other baselines**. We include two recently proposed approaches, including MMA and DMN.

| Step | Flowers102 | | Food101 | | FGVCAircraft | | StanfordCars | | Average | |
|---|---|---|---|---|---|---|---|---|---|---|
| | Acc | Time | Acc | Time | Acc | Time | Acc | Time | Acc | Time |
| 25 | 72.24 | 10min | 91.18 | 6min | 27.23 | 8min | 73.55 | 13min | 66.05 | 9min |
| 50 | 72.85 | 15min | 91.96 | 11min | **28.41** | 14min | 74.67 | 23min | 66.97 | 16min |
| 75 | 72.81 | 22min | **92.28** | 18min | 28.19 | 24min | 74.82 | 31min | 67.02 | 24min |
| 100 | **72.99** | 28min | 92.12 | 26min | 28.30 | 31min | **74.96** | 40min | **67.09** | 31min |

Table 19: **The performance of SAS and diffusion time with different number of diffusion steps.**

datasets, proving it to be an effective plug-and-play method that can be generalized to more complex modalities and tasks.

### B.16    EVALUATION ON MORE BASELINES

For completeness, here we evaluate our method on the two recently proposed works. Specifically, we select MMA (Yang et al., 2024) and DMN (Zhang et al., 2024b). For the former, we train the newly introduced adapters in the deep layers that bridge the text and image representations, following their setting and implementation. For the latter, we optimize its memory projection functions and incorporate both the static and dynamic memory networks, which is the strongest variant according to their paper. We select the base-to-new generalization task, as illustrated in Section 4, and record the new category accuracy, which directly reflects the generalization performance. As shown in Table 18, SAS consistently improves performance on both methods, demonstrating its complementarity.

### B.17    ABLATION ON DIFFUSION STEPS

Considering the computations introduced by diffusion in generating images, here we perform an ablation study on the efficiency of diffusion inference. Specifically, we vary the number of diffusion steps, which is the key hyperparameter influencing the inference time. Intuitively, fewer steps are more efficient yet yield lower image quality, while more steps ensure image fidelity but require more computation. We select CoCoOp as the baseline and record the new category accuracy on base-to-new generalization. As shown in Table 19, by default, we use 100 steps throughout the paper as described in Section A.4, which requires an average of 31 minutes to generate images per dataset. Here we try fewer steps, such as 50, and observe that the time required for diffusion nearly halves ($31\text{min} \rightarrow 16\text{min}$) with minimal degradation in performance ($67.09 \rightarrow 66.97$). However, while the number of steps is further reduced to 25, there is a dramatic performance drop ($66.97 \rightarrow 66.05$), possibly due to the decline in image quality. This suggests we may safely adjust the number of steps from 100 to 50, which halves the required time with minimal accuracy loss, significantly improving the efficiency of SAS.

| Method | ImageNet | Caltech101 | OxfordPets | StanfordCars | DTD | EuroSAT | ImageNet-A | Average |
|---|---|---|---|---|---|---|---|---|
| CLIP | 66.54 | 94.62 | 90.41 | 64.69 | 44.84 | 47.50 | 49.32 | 65.42 |
| DCLIP | 68.52 | 95.48 | 91.88 | 65.70 | 45.52 | 49.08 | 49.88 | 66.58 |
| DCLIP - SA | 67.67 | 94.76 | 91.31 | 64.79 | 45.06 | 47.92 | 49.30 | 65.83 |
| CuPL | 69.99 | 96.51 | 92.62 | 66.91 | 47.32 | 50.33 | 50.14 | 67.69 |
| CuPL - SA | 68.70 | 95.89 | 92.38 | 65.66 | 46.06 | 49.58 | 49.28 | 66.79 |

Table 20: **The zero-shot accuracy before and after removing spurious attributes.** The model is evaluated on 2 generic datasets (ImageNet (Deng et al., 2009), Caltech101 (Fei-Fei et al., 2004)), 2 fine-grained datasets (OxfordPets (Parkhi et al., 2012), StanfordCars (Krause et al., 2013)), 2 specialized datasets (DTD (Cimpoi et al., 2014), EuroSAT (Helber et al., 2019)) and 1 adversarial dataset (ImageNet-A (Hendrycks et al., 2021b)).

## C   FURTHER EXPLORATION

### C.1   SPURIOUS ATTRIBUTES FOR ZERO-SHOT RECOGNITION

The primary takeaway of this paper is the unbalanced treatment of various semantic attributes by VLMs, which extends beyond the generalization task and suggests that the language encoder of VLMs may allocate distinct attention to different tokens. We examine a typical example: zero-shot recognition, where attributes are directly utilized to make predictions without training. We consider three baselines: CLIP (Radford et al., 2021), DCLIP (Menon & Vondrick, 2023), and CuPL (Pratt et al., 2023). The latter two employ LLMs to generate attributes and enhance zero-shot accuracy. In a manner similar to the previous motivational study, we remove identified spurious attributes from the existing baselines and record the accuracy before and after this intervention. Table 20 presents the results before and after removal. We observe a significant drop in accuracy for the baselines (from $63.23\%$ to $62.49\%$ for DCLIP and from $64.34\%$ to $63.45\%$ for CuPL), with DCLIP almost reverting to the performance of vanilla CLIP ($62.49\%$ vs. $62.07\%$). This indicates that 1) similar to the generalization task, zero-shot recognition is also dominated by spurious attributes, nearly ignoring the presence of other generated attributes; and 2) spurious attributes, in a sense, improve zero-shot performance on natural datasets by scaling up the model's inherent bias.

### C.2   SELECTIVE OPTIMIZATION TRICK

`SAS` introduces a subsidiary task that includes constructed pseudo categories and auxiliary learning objectives. With an increasing number of spurious attributes, a large number of pseudo categories are introduced, significantly increasing computational costs. To tackle this challenge, we introduce a strategy that selectively optimizes partial target categories with a heavy bias towards spurious attributes. In other words, we only mitigate the influence of spurious attributes on categories that overly rely on them. To identify these categories, we propose Spurious Correlation Ratio (SCR). SCR is calculated as the ratio of the average weights of spurious attributes to the average weights of all attributes, as exemplified in the rightmost column of Table 6. A higher SCR indicates that the prediction of the corresponding category relies more on spurious attributes. In implementing this trick, we empirically select only the top 10% of categories ranked by SCR for optimization. To verify the trick, we choose two time-intensive baselines, CoCoOp (Zhou et al., 2022b) and PromptSRC (Khattak et al., 2023b), for comparison. CoCoOp's training is slow due to its instance-conditioned mechanism, while PromptSRC adds three extra learning objectives to the original cross-entropy loss. To emphasize the results, we conduct evaluation on the base-to-new generalization task using ImageNet (Deng et al., 2009) and record both training time and harmonic mean accuracy. Table 5 in the main paper illustrates the trade-off between effectiveness and efficiency with `SAS` and the proposed trick. It is evident that integrated with selective optimization, the required time is significantly reduced compared to the original `SAS`. For instance, on PromptSRC, it only adds 9 minutes of training time while preserving most of the performance gains.

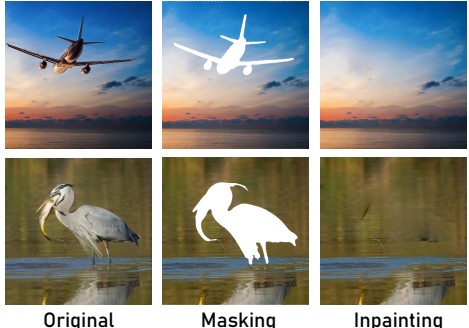

Original          Masking          Inpainting

Figure 10: **The constructed categories with masking and inpainting.** For the former, we directly mask the primary object with SAM (Kirillov et al., 2023), whereas for the latter, we further use RePaint (Lugmayr et al., 2022) to fill in the missing parts. We compare the performance between the pseudo categories constructed with masking, inpainting and our synthesis method.

## C.3 VARIANTS OF SAS

In the main paper, SAS primarily constructs pseudo categories using synthetic or pre-trained data, which has proven effective. Here, we consider two simple yet direct variants: 1) instead of utilizing spurious attributes to create new data, we directly mask the main object in the original images and use these as the corresponding pseudo categories, termed SAS-masking; 2) upon masking, we fill the masked area through in-painting, termed SAS-inpainting. Fig. 10 displays some example images of pseudo categories by these two variants. The motivation here is to enhance VLMs' awareness of core attributes by contrasting target categories with their corresponding images that lack main objects. We refer to the original approach as SAS-synthesis, where pseudo categories are constructed with SD-synthesized images. Table 21 presents the performance of the three methods, showing that both variants perform worse than SAS-synthesis. We speculate that this is because 1) masking or in-painting significantly reduces image fidelity; and 2) this approach introduces excessive noise attributes, thereby forming a new set of spurious attributes for VLMs to learn.

## C.4 NON-SEMANTIC SPURIOUS ATTRIBUTE

In the evaluated datasets, including previous work on measuring group robustness (Zhang & Ré, 2022; Adila et al., 2023), most spurious attributes are semantically related, wherein the attribute and label exhibit a natural association, *e.g.*, water and water bird. In this study, we extend our exploration to non-semantic attributes, where the association between the attribute and label is artificially constructed. We implement a straightforward color-shifting experiment using ColoredM-NIST. This dataset comprises 10 classes, each representing a digit; however, instead of the standard black background in MNIST, each digit class features a distinctly colored background. Each color demonstrates a strong spurious correlation with its corresponding digit, effectively serving as a spurious attribute. Fig. 11 illustrates examples from ColoredMNIST. We employ GPT-4V to identify these non-semantic spurious attributes, resulting in descriptors such as green background and pure yellow background. We evaluate SAS on the test set of ColoredMNIST, where the color backgrounds are randomized across labels. As shown in Table 22, SAS significantly enhances VLMs'

| Method | CoCoOp | MaPLe | PromptSRC |
|---|---|---|---|
| SAS-masking | 71.08 | 74.53 | 75.39 |
| SAS-inpainting | 72.95 | 76.81 | 77.04 |
| SAS-synthesis | **73.50** | **77.69** | **77.88** |

Table 21: **The evaluation on base-to-new generalization with two SAS variants.**

| Method | CoCoOp | MaPLe | PromptSRC |
|--------|--------|-------|-----------|
| w/o SAS | 72.47 | 75.89 | 74.08 |
| w/ SAS | 84.70 | 88.27 | 87.48 |

Table 22: **The evaluation on ColoredMNIST with and without `SAS`.**

robustness to color shifting, indicating that MLLMs may capture non-semantic attributes in images, and `SAS` effectively leverages these attributes to improve generalization.

## C.5 LIMITATION AND FAILURE CASES

In `SAP`, the primary limitation stems from the necessity of having available images. Previous approaches to generating visual attributes only require textual information, *e.g.*, category names. The underlying assumption is that the generated attributes would be dataset-agnostic. For example, attributes like headlights, doors, or wheels for the category vehicle are assumed to be consistent across datasets. However, spurious attributes do not adhere to this assumption; they are contingent on the specific characteristics of the dataset. For instance, vehicle images in different datasets might be taken on a highway or in a parking lot, resulting in vastly different spurious attributes. This highlights the need for visual information from the dataset itself to accurately identify spurious attributes.

For `SAS`, the main concern still lies in efficiency. While the use of synthetic or pre-training images has been employed to address data scarcity in many recent works, such as SuS-X (Udandarao et al., 2022) and Real-Prompt (Parashar et al., 2024), these methods inevitably introduce additional computational overhead. The inference of Stable Diffusion (Rombach et al., 2022), relative to its large data requirements, is not particularly fast, and retrieval requires finding top-k matches from a huge pre-training dataset (Schuhmann et al., 2022), both of which have efficiency bottlenecks. While selective optimization tricks can minimize computational burdens as much as possible, they come at the cost of accuracy.

## C.6 MORE RELATED WORKS

**Retrieval-Augmented Generation**. RAG is proposed essentially to address the insufficiency or lack of desired data. For example, Long et al. (2022) improves long-tail recognition performance by retrieving text representations for tail classes. Similarly, Parashar et al. (2024) enhances VLMs' tail accuracy by identifying and retrieving high-frequency text synonyms corresponding to tail names from the training set. Furthermore, Udandarao et al. (2022) mitigates data sparsity issues by retrieving external images through class names for data augmentation. Sharing motivations with previous work, we construct pseudo categories featuring spurious attributes through retrieval, thereby enhancing the model's robustness to these attributes. Nevertheless, beyond retrieval, we also explore data synthesis. In Section B.5, we compare the performance of our method using synthesized and retrieved data, empirically concluding that synthesized data yields greater accuracy gains. Compared to retrieval, synthesis can offer more tailored and precise scenarios and objects, which may be more suitable for our method given the diverse identified attributes.

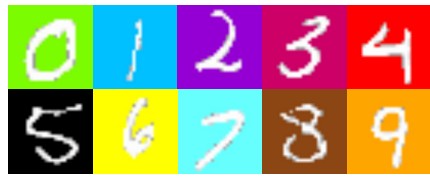

Figure 11: **ColoredMNIST.**

## D  BROADER SOCIETAL IMPACTS

Our work has positive societal impacts. As illustrated in Fig. 5 of the main paper, VLMs may exhibit bias by associating harmful spurious attributes with target categories. For instance, when recognizing street sign, VLMs often rely excessively on concepts like street and road. This non-robust visual perception may lead to severe consequences in real-world applications, particularly in autonomous driving. The introduction of SAP can effectively identify such harmful attributes and even create a spurious attribute pool for specific applications, helping to determine situations where performance is compromised. Meanwhile, SAS provides an effective approach to suppress the influence of spurious attributes in VLMs, significantly enhancing the model's robustness against these attributes, including protected ones such as gender and race. Currently, we have not identified negative societal impacts of this work. However, due to objective factors, such as the availability of datasets and baselines' code, this will need to be further discussed in the future.

## E  SUPPLEMENTARY RESULTS

### E.1  CONSTRUCTED IMAGES

In Fig. 12, we provide more constructed images by SAS, with Stable Diffusion and retrieval from LAION-5B, respectively.

### E.2  SPURIOUS ATTRIBUTE STATISTICS

Here, we present the spurious attribute statistics for the evaluated datasets. Specifically, we report the proportion of images containing one or more spurious attributes identified by SAS across 11 datasets, as shown in Table 23. The data reveals that for most datasets, over 50% of images contain spurious attributes, highlighting the biased nature of these datasets and the consequent spurious correlations learned by VLMs.

### E.3  MOTIVATIONAL RESULTS

Given the enhanced generalization performance of VLMs before and after removing spurious attributes in Table 1 of the main paper, to further illustrate the impact of spurious attributes, here we present the improvement of the models on the counter group in Table 24. It can be observed that the accuracy of VLMs on the counter group shows a more significant improvement, up to 9% on the unseen categories.

### E.4  NUMERICAL MAIN RESULTS

Here we quantitatively demonstrate the main results as depicted in Fig. 3 of the main paper. Table 25 and Table 26 present the numerical results of base-to-new generalization, cross-dataset transfer, and domain generalization, respectively.

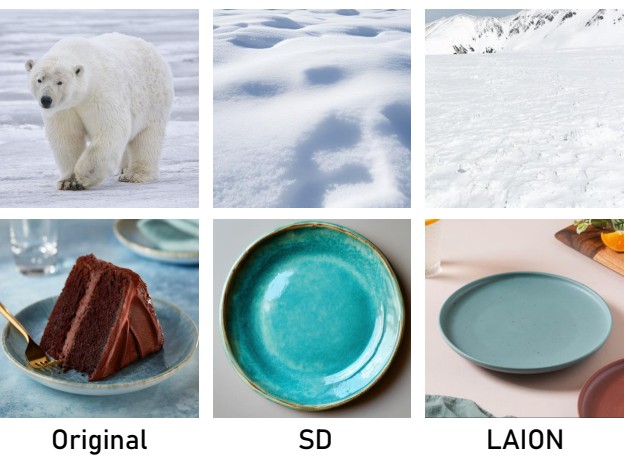

Figure 12: **More examples of generated and retrieved images with Stable Diffusion (SD) and LAION, respectively.** The category above is polar bear, with the primary spurious attribute being *snow-covered ground*. The category below is chocolate cake, where one of the spurious attributes is the *dinner plate*.

| Dataset | Images with spurious attributes (%) |
|---|---|
| ImageNet | 62.48 |
| Caltech101 | 58.22 |
| OxfordPets | 73.54 |
| StanfordCars | 69.92 |
| Flowers102 | 63.50 |
| Food101 | 54.59 |
| FGVCAircraft | 47.97 |
| SUN397 | 52.20 |
| DTD | 42.68 |
| EuroSAT | 47.90 |
| UCF101 | 71.57 |

Table 23: **The proportion of images containing one or more spurious attributes of 11 datasets.**

| Method | FGVCAircraft | | | SUN397 | | | Flowers102 | | | DTD | | | Average | | |
|---|---|---|---|---|---|---|---|---|---|---|---|---|---|---|---|
| | Base | New | SR | Base | New | SR | Base | New | SR | Base | New | SR | Base | New | SR |
| CPL | 29.25 | 24.80 | 5.43 | 63.47 | 54.74 | 6.61 | 76.98 | 60.77 | 5.71 | 68.93 | 43.14 | 5.13 | 59.66 | 45.86 | 5.72 |
| CPL - SA | **32.42** | **31.34** | — | 64.81 | 60.35 | — | **78.35** | **69.80** | — | 71.26 | **52.65** | — | 61.71 | **53.54** | — |
| ArGue | 27.24 | 21.92 | 5.13 | 65.23 | 57.44 | 6.45 | 72.26 | 59.32 | 6.69 | 70.77 | 40.12 | 5.97 | 58.87 | 44.70 | 6.06 |
| ArGue* | 27.92 | 23.83 | 4.86 | 65.99 | 58.21 | 6.11 | 72.87 | 60.49 | 6.44 | 71.29 | 41.28 | 5.62 | 61.27 | 45.95 | 5.76 |
| ArGue - SA | 30.71 | 29.20 | — | **67.90** | **63.68** | — | 74.68 | 66.45 | — | **73.70** | 50.19 | — | **61.75** | 52.38 | — |

Table 24: **The results on the counter group in base-to-new generalization before and after removing spurious attributes (SA) from the pool.** We extract the counter group for both the base and new categories where spurious cues are removed. It can be observed that the accuracy of VLMs improves after removing spurious attributes in this context.

| Method | ImageNet | | Caltech | | OxfordPets | | Cars | | Flowers | | Food | | Aircraft | | SUN | | DTD | | EuroSAT | | UCF | |
|---|---|---|---|---|---|---|---|---|---|---|---|---|---|---|---|---|---|---|---|---|---|---|
| | Base | New | Base | New | Base | New | Base | New | Base | New | Base | New | Base | New | Base | New | Base | New | Base | New | Base | New |
| CLIP | 72.43 | 68.14 | 96.84 | 94.0 | 91.17 | 97.26 | 63.37 | 74.89 | 72.08 | 77.80 | 90.10 | 91.22 | 27.19 | 36.29 | 69.36 | 75.35 | 53.24 | 59.90 | 56.48 | 64.05 | 70.53 | 77.50 |
| CoCoOp | 75.98 | 70.43 | 97.96 | 93.81 | 95.20 | 97.69 | 70.49 | 73.59 | 94.87 | 71.75 | 90.70 | 91.29 | 33.41 | 23.71 | 79.74 | 76.86 | 77.01 | 56.00 | 87.49 | 60.04 | 82.33 | 73.45 |
| + SAS | 76.40 | 71.38 | 97.33 | 94.62 | 95.73 | 97.80 | 70.70 | 74.96 | 95.17 | 72.99 | 90.58 | 92.12 | 34.06 | 28.30 | 80.40 | 79.67 | 76.87 | 57.73 | 87.77 | 63.61 | 82.57 | 75.66 |
| KgCoOp | 75.83 | 69.96 | 97.72 | 94.39 | 94.65 | 97.76 | 71.76 | 75.04 | 95.00 | 74.73 | 90.50 | 91.70 | 36.21 | 33.55 | 80.29 | 76.53 | 77.55 | 54.99 | 85.64 | 64.34 | 82.89 | 76.67 |
| + SAS | 75.78 | 71.24 | 97.96 | 95.40 | 95.43 | 98.67 | 71.46 | 75.93 | 95.51 | 75.86 | 90.69 | 92.03 | 36.93 | 36.20 | 80.80 | 79.02 | 77.89 | 60.30 | 85.78 | 73.73 | 83.38 | 78.94 |
| MaPLe | 76.66 | 70.54 | 97.74 | 94.36 | 95.43 | 97.76 | 72.94 | 74.00 | 95.92 | 72.46 | 90.71 | 92.05 | 37.44 | 35.61 | 80.82 | 78.70 | 80.36 | 59.18 | 94.07 | 73.23 | 83.00 | 78.66 |
| + SAS | 76.69 | 70.82 | 97.92 | 95.35 | 95.88 | 98.47 | 73.16 | 75.46 | 95.93 | 76.74 | 91.41 | 92.47 | 37.87 | 39.68 | 81.30 | 80.72 | 80.78 | 63.21 | 94.38 | 78.45 | 82.89 | 80.24 |
| PromptSRC | 77.60 | 70.73 | 98.10 | 94.03 | 95.33 | 97.30 | 78.27 | 74.97 | 98.07 | 76.50 | 90.67 | 91.53 | 42.73 | 37.87 | 82.67 | 78.47 | 83.37 | 62.97 | 92.90 | 73.90 | 87.10 | 78.80 |
| + SAS | 77.48 | 71.48 | 98.52 | 95.20 | 95.92 | 98.50 | 78.62 | 75.24 | 98.45 | 79.11 | 90.99 | 92.43 | 42.64 | 40.16 | 83.23 | 80.80 | 83.94 | 63.46 | 93.35 | 76.68 | **87.66** | 81.55 |
| LASP | 76.25 | 71.17 | 98.17 | 94.33 | 95.73 | 97.87 | 75.23 | 71.77 | 97.17 | 73.53 | 91.20 | 91.90 | 38.05 | 33.20 | 80.70 | 79.30 | 81.10 | 62.57 | 95.00 | 83.37 | 85.53 | 78.20 |
| + SAS | 76.62 | 72.45 | 98.44 | 95.27 | 96.00 | 98.58 | 76.29 | 72.85 | 96.98 | 75.30 | 92.07 | 92.65 | 38.58 | 33.97 | 80.99 | 80.42 | 81.45 | 64.11 | **95.61** | 83.69 | 85.70 | 79.19 |
| TCP | 77.27 | 69.87 | 98.23 | 94.67 | 94.67 | 97.20 | 80.80 | 74.13 | 97.73 | 75.57 | 90.57 | 91.37 | 41.97 | 34.43 | 82.63 | 78.20 | 82.77 | 58.07 | 91.63 | 74.73 | 87.13 | 80.77 |
| + SAS | 77.89 | 70.53 | 98.42 | 95.40 | 95.48 | 98.23 | 80.68 | 75.80 | 98.43 | 75.48 | 91.18 | 92.70 | 42.58 | 35.10 | **83.41** | 79.27 | 83.10 | 60.67 | 92.01 | 75.34 | 87.50 | 81.62 |
| CLIP-Adapter | 77.18 | 70.25 | 97.52 | 93.48 | 95.18 | 96.43 | 77.43 | 72.64 | 96.83 | 71.75 | 90.98 | 90.55 | 41.89 | 33.10 | 80.73 | 77.98 | 82.17 | 58.72 | 93.34 | 71.84 | 86.49 | 77.38 |
| + SAS | 77.09 | 71.98 | 98.10 | 95.55 | 95.72 | 97.70 | 77.63 | 75.46 | 97.59 | 73.13 | 90.74 | 91.38 | 41.88 | 36.50 | 81.42 | 80.15 | 82.84 | 61.27 | 93.99 | 73.87 | 86.87 | 78.51 |
| Tip-Adapter | 78.04 | 71.96 | 98.68 | 94.17 | 96.21 | 98.57 | 80.79 | 73.66 | 98.73 | 74.36 | 92.62 | 91.01 | **43.34** | 35.73 | 81.77 | 79.27 | 84.58 | 59.91 | 94.82 | 74.81 | 86.78 | 78.94 |
| + SAS | 77.89 | 72.58 | **98.89** | 95.72 | 96.65 | 98.43 | **80.84** | 75.42 | **98.81** | 76.73 | **92.84** | 92.81 | 43.27 | 38.31 | 82.36 | 80.13 | **84.77** | 63.38 | 95.51 | 77.54 | 87.42 | 79.77 |
| ArGue | 76.95 | 71.86 | 98.63 | 94.70 | 96.23 | 98.59 | 75.06 | 74.18 | 98.62 | 77.96 | 91.42 | 92.40 | 41.29 | 38.80 | 81.89 | 80.48 | 80.33 | 67.03 | 95.10 | 90.68 | 86.00 | 79.43 |
| + SAP | 77.32 | 72.04 | 98.57 | 95.12 | 96.34 | 98.86 | 75.72 | 74.90 | 98.66 | 78.78 | 91.54 | 92.63 | 41.86 | 39.65 | 82.43 | 81.78 | 80.87 | 68.36 | 95.46 | 91.51 | 86.59 | 80.28 |
| + SAS | 77.59 | 72.36 | 98.69 | 95.88 | 96.52 | 98.75 | 76.24 | 75.51 | 98.74 | 79.65 | 91.81 | 93.42 | 42.39 | 40.84 | 82.71 | 82.21 | 81.35 | **69.73** | 95.41 | **92.47** | 86.67 | 81.73 |
| MAP | 76.60 | 70.60 | 98.30 | 93.80 | 95.43 | 96.90 | 76.70 | 73.73 | 97.57 | 75.23 | 90.30 | 89.30 | 41.63 | 36.43 | 82.33 | 76.30 | 82.63 | 66.23 | 92.13 | 76.10 | 86.67 | 78.77 |
| + SAP | 76.73 | 71.17 | 98.21 | 94.32 | 95.79 | 98.09 | 77.34 | 74.10 | 97.85 | 77.55 | 90.60 | 90.76 | 42.05 | 37.72 | 82.15 | 78.05 | 82.77 | 67.61 | 92.53 | 77.22 | 87.09 | 79.42 |
| + SAS | 76.79 | 72.25 | 98.53 | 94.52 | 96.19 | 98.83 | 77.70 | 75.80 | 97.79 | 79.99 | 90.89 | 91.63 | 42.33 | 39.32 | 82.74 | 78.47 | 82.87 | 68.44 | 93.30 | 78.21 | 87.44 | 80.85 |
| CPL | 78.74 | 72.03 | 98.35 | 95.13 | 95.86 | 98.21 | 79.31 | 76.65 | 98.07 | 80.43 | 91.92 | 93.87 | 42.27 | 38.85 | 81.88 | 79.65 | 80.92 | 62.27 | 94.18 | 81.05 | 86.73 | 80.17 |
| + SAP | 78.76 | 72.64 | 98.67 | 95.72 | 96.31 | **98.87** | 79.24 | 78.12 | 98.37 | 82.51 | 92.19 | 94.63 | 42.15 | 40.92 | 82.16 | 81.62 | 82.21 | 65.72 | 94.47 | 83.55 | 86.51 | 80.91 |
| + SAS | **78.82** | **73.49** | 98.59 | **95.98** | **96.76** | 98.82 | 79.77 | **80.35** | 98.71 | **83.46** | 92.26 | **95.45** | 42.61 | **41.72** | 82.11 | **83.17** | 83.00 | 67.89 | 94.75 | 87.07 | 87.21 | **82.22** |

Table 25: **The numerical results on base-to-new generalization.**

| | Source | Cross-dataset Transfer Target | | | | | | | | | | Domain Generalization Target | | | |
|---|---|---|---|---|---|---|---|---|---|---|---|---|---|---|---|
| | ImageNet | Caltech | Pets | Cars | Flowers | Food | Aircraft | SUN | DTD | EuroSAT | UCF | ImageNet-V | ImageNet-S | ImageNet-A | ImageNet-R |
| CLIP | 66.54 | 94.62 | 90.41 | 64.69 | 70.30 | 85.63 | 23.73 | 66.12 | 44.84 | 47.50 | 67.42 | 63.20 | 48.35 | 49.32 | 76.57 |
| CoCoOp | 71.02 | 94.43 | 90.14 | 65.32 | 71.88 | 86.06 | 22.94 | 67.36 | 45.73 | 45.37 | 68.21 | 64.07 | 48.75 | 50.63 | 76.18 |
| +SAS | 71.35 | 95.59 | 90.84 | 66.76 | 72.47 | 86.34 | 23.81 | 68.99 | 47.94 | 46.90 | 70.84 | 64.97 | 49.56 | 51.61 | 77.31 |
| KgCoOp | 70.66 | 93.92 | 89.83 | 65.41 | 70.01 | 86.36 | 22.51 | 66.16 | 46.35 | 46.04 | 68.50 | 64.10 | 48.97 | 50.69 | 76.70 |
| +SAS | 70.90 | 94.33 | 89.68 | 67.82 | 71.13 | 88.91 | 24.60 | 67.47 | 47.72 | 48.22 | 68.52 | 64.53 | 49.72 | 51.70 | 77.22 |
| MaPLe | 70.72 | 93.53 | 90.49 | 65.57 | 72.23 | 86.20 | 24.74 | 67.01 | 46.49 | 48.06 | 68.69 | 64.07 | 49.15 | 50.90 | 76.98 |
| +SAS | 71.21 | 93.61 | 91.76 | 67.53 | 73.60 | 87.58 | 24.54 | 67.69 | 47.98 | 48.17 | 71.96 | 63.98 | 50.74 | 51.57 | 77.25 |
| PromptSRC | 71.27 | 93.60 | 90.25 | 65.70 | 70.25 | 86.15 | 23.90 | 67.10 | 46.87 | 45.50 | 68.75 | 64.35 | 49.55 | 50.90 | 77.80 |
| +SAS | 71.53 | 93.25 | 92.60 | 66.44 | 70.13 | 88.19 | 25.05 | 67.87 | 47.22 | 45.50 | 68.99 | 64.07 | 50.40 | 51.52 | **78.98** |
| LASP | 71.34 | 93.65 | 91.83 | 67.29 | 70.82 | 88.54 | 28.60 | 65.75 | 54.83 | 43.65 | 69.23 | 64.04 | 47.93 | 49.11 | 75.36 |
| +SAS | 71.62 | 94.62 | 92.98 | 68.89 | 71.18 | 89.89 | 29.68 | 68.47 | 55.74 | 45.80 | 71.63 | 65.24 | 47.91 | 50.80 | 77.08 |
| TCP | 71.40 | 93.97 | 91.25 | 64.69 | 71.21 | 86.69 | 23.45 | 67.15 | 44.35 | 51.45 | 68.73 | 64.60 | 49.50 | 51.20 | 76.73 |
| +SAS | 71.73 | 94.73 | 92.60 | 66.54 | 71.44 | 87.81 | 24.80 | 68.94 | 45.15 | 52.93 | 70.31 | 65.62 | 50.79 | **52.94** | 78.82 |
| CLIP-Adapter | 72.35 | 93.06 | 90.76 | 63.17 | 69.23 | 85.13 | 20.54 | 65.57 | 43.27 | 49.64 | 66.33 | 62.91 | 49.15 | 51.74 | 76.81 |
| +SAS | 72.53 | 93.12 | 91.72 | 66.65 | 69.18 | 88.10 | 22.27 | 66.60 | 45.69 | 50.38 | 69.80 | 64.50 | 49.70 | 52.39 | 77.75 |
| Tip-Adapter | 72.53 | 95.71 | 93.12 | 66.61 | 68.83 | 89.22 | 23.63 | 68.32 | 47.31 | 53.40 | 68.15 | 63.30 | 49.26 | 50.18 | 76.70 |
| +SAS | 72.81 | 95.49 | **94.88** | 67.80 | 68.46 | 91.77 | 25.00 | 69.46 | 49.55 | **54.33** | 68.94 | 64.21 | 50.34 | 50.89 | 77.93 |
| ArGue | 71.84 | 94.20 | 92.66 | 70.70 | 71.29 | 91.64 | 28.28 | 70.51 | 55.37 | 45.76 | 71.97 | 65.02 | 49.25 | 51.47 | 76.96 |
| +SAP | 72.14 | 95.74 | 93.75 | 71.80 | 72.48 | 91.87 | 28.53 | 70.88 | 56.54 | 46.86 | 72.96 | 65.47 | 49.94 | 52.48 | 77.38 |
| +SAS | 72.28 | 95.67 | 94.29 | **72.72** | 74.63 | **92.53** | 29.10 | **71.96** | **57.40** | 48.22 | **73.82** | 66.12 | 49.90 | 52.85 | 77.90 |
| MAP | 71.60 | 93.93 | 90.80 | 63.00 | 68.40 | 86.07 | 24.87 | 68.10 | 51.87 | 42.63 | 68.73 | 64.47 | 49.07 | 51.07 | 77.37 |
| +SAP | 71.93 | 95.40 | 92.63 | 64.50 | 68.13 | 87.18 | 26.80 | 69.99 | 51.35 | 44.10 | 70.50 | 65.06 | 49.88 | 51.64 | 77.34 |
| +SAS | 72.21 | 95.82 | 93.73 | 66.69 | 68.46 | 88.11 | 28.62 | 70.29 | 51.91 | 45.73 | 71.59 | 66.14 | 50.78 | 52.19 | 77.70 |
| CPL | 73.53 | 95.52 | 91.64 | 66.17 | 73.35 | 87.68 | 27.36 | 68.24 | 48.96 | 51.25 | 70.52 | 65.24 | 50.84 | 52.10 | 76.76 |
| +SAP | 73.75 | **95.83** | 92.92 | 66.69 | 74.32 | 88.33 | 29.58 | 69.64 | 49.81 | 52.72 | 71.35 | **66.45** | 51.93 | 52.61 | 77.74 |
| +SAS | **73.94** | 95.74 | 93.67 | 67.22 | **75.67** | 89.49 | **30.55** | 70.26 | 49.91 | 54.29 | 72.48 | 66.38 | **52.95** | 52.81 | 78.31 |

Table 26: **The numerical results on cross-dataset transfer and domain generalization.**

