# OpenReview forum: "Black Sheep in the Herd: Playing with Spuriously Correlated Attributes for Vision-Language Recognition"
_ICLR.cc/2025/Conference — ICLR 2025 Poster_

### Official Review · Reviewer_rex7 · 2024-10-30

**Soundness:** 3
**Presentation:** 4
**Contribution:** 3
**Rating:** 8
**Confidence:** 2

**Summary:**

This paper proposes a new adaptation strategy for improving the generalization of attribute-based VLM recognition methods. The authors assess the detrimental impact of spurious attribute on adapting to novel domains, and propose spurious attribute probing and shielding strategies to mitigate this influence. By learning a subsidiary task to discriminate target categories and spurious attribute categories, the adaptation performance on out-of-distribution data is notably improved.

**Strengths:**

1.The motivation of this paper appears interesting to me. The authors quantitively identify the influence of spurious attributes,  a key problem for out-of-distribution visual recognition.

2.The idea of learning pseudo category to mitigate the impact of spurious attribute is novel. The proposed SAS and SAP bring consistent improvement across 10 baselines, demonstrating the method’s effectiveness.

3.The paper is well-written, well-organized and easy to follow.

**Weaknesses:**

1.During fine-tuning, differentiating each category from the synthetic pseudo category is time-consuming and tedious. Although the authors apply selective optimization to maintain overall performance，they have not fundamentally addressed the prolonged optimization time for each category.

2.Only three visualization figs are presented in Figure 5, and the results can not effectively demonstrate the removal of attention to spurious attribute features, e.g., Figure 5 (b).

3.During training, an additional module generates negative samples to form a strong contrast from the target categories. There’s a possibility that the performance improvement is primarily attributed to data augmentation, as the authors also mentioned in the paper. However, the authors do not compare the performance when generating an equivalent number of positive samples for each target category.

**Questions:**

1.It is advisable to provide more visualization examples in Figure 5 or in supplementary materials.

2.Can you provide an ablation study of the performance when generating an equivalent number of positive samples for each category?

---

> ### Author Response · Authors · 2024-11-19
> **Response to Reviewer rex7 by Authors (1/2)**
>
> We sincerely appreciate your valuable comments and recognition of our work. Below are our responses to your concerns.
>
> >**W1: The optimization efficiency of the proposed method**
>
> In Table 5 of the main paper, we demonstrate the optimization efficiency of our method and the effectiveness of the selective optimization trick. As shown, with the same number of epochs, applying selective optimization allows SAS to train on ImageNet with only approximately 10 additional minutes compared to the baselines. Meanwhile, our method does not introduce any extra computations during testing.
>
> Here, we present the optimization time costs for SAS and selective optimization across other datasets. The time is determined as the runtime of the training script, which is based on the implementation of CoOp [A].
>
> | Method             | Flowers102 | Food101 | FGVCAircraft | StanfordCars | Average Time |
> |--------------------|:----------:|:------:|:------------:|:------------:|:------------:|
> | CoCoOp [B]             |   10m48s   | 18m34s |     8m02s    |    12m46s    |    12m32s    |
> | + SAS              |   13m14s   | 24m07s |    13m36s    |    17m08s    |    17m01s    |
> | + selective trick  |   11m03s   | 20m45s |    10m23s    |    14m55s    |    14m16s    |
> | PromptSRC [C]          |    6m25s   | 15m09s |     5m44s    |     9m36s    |     9m13s    |
> | + SAS              |    8m26s   | 18m21s |     7m11s    |    12m04s    |    11m30s    |
> |  + selective trick |    6m58s   | 16m22s |     6m35s    |    10m20s    |    10m03s    |
>
> As shown in the table above, for most datasets, the integration of SAS only increases the training time by approximately 3 to 5 minutes, while selective optimization further reduces this time to a negligible amount. In fact, the selective optimization trick is proposed to address large-scale datasets, such as ImageNet, which contains 1000 categories. For regular datasets (~100 categories), the time consumption of SAS is fully acceptable. In the revised paper, we have provided a detailed analysis of the optimization efficiency of our method in Section B.14 of the Appendix (highlighted in blue).
>
> >**W2: More visualization examples of the proposed method**
>
> In Section 5 of the main paper, we visualize the saliency map of example images with and without SAS. For the chocolate cake in (a), SAS succesfully mitigates VLMs' reliance on typical spurious attributes, such as plates and utensils. In (b), as mentioned by the reviewer, for the personal laptop, SAS significantly reduces attention to likely its most common acommpanying object, i.e., the mouse. In (c), for street sign recognition, SAS removes the model's bias towards the road, which is particularly beneficial for autonomous driving. It is worth mentioning that the saliency maps here are used for a qualitative understanding of the method, as they are not perfect indicators of localization.
>
> However, we appreciate the reviewer's suggestion that more visualization examples are expected to demonstrate the effectiveness of SAS. We have revised the paper and included six additional examples in Section B.12 of the Appendix (highlighted in blue). As shown in Figure 9, SAS consistently mitigates the impact of spurious cues across various categories. For instance, for the tree frog, SAS reduces the VLM's dependence on tree branches, while for the airliner, the model no longer focuses on the sky or clouds, which are typicall spuriously correlated with airliners. Similarly, for the polar bear, where the snow-covered ground is a common spurious attribute, SAS effectively alleviates the model's bias towards it.
>
> **Reference**
>
> [A] Learning to Prompt for Vision-Language Models, IJCV2022
>
> [B] Conditional Prompt Learning for Vision-Language Models, CVPR2022
>
> [C] Self-regulating Prompts: Foundational Model Adaptation without Forgetting, ICCV2023

---

> ### Author Response · Authors · 2024-11-19
> **Response to Reviewer rex7 by Authors (2/2)**
>
> >**W3: Performance gains from the proposed method or data augmentation**
>
> Our proposed method, SAS enhances model performance by teaching it to distinguish between main objects and spurious attributes through the construction of pseudo categories that feature the latter, i.e., negative samples as mentioned by the reviewer. Therefore, the performance gains should stem from increased model robustness to spurious cues rather than from the vanilla data augmentation. In Section 4.2 of the main paper, we conduct a simple ablation study to verify the contribution of spurious attributes on our method by adjusting $\gamma$.
>
> Here, as suggested by the reviewer, we further conduct an ablation study to compare the performance improvements brought by extra data and our proposed method to prove this point. Specifically, along with our proposed method, we design two baselines. In the first baseline, we involve additional data directly from the original dataset featuring the main objects, extending the training data from 16 shots to 32 shots (32-shot main). In the second baseline, we consider additional data generated by pseudo categories that are the same as the main categories, i.e., positive samples as mentioned by the reviewer (16-shot main + 16-shot positive). In contrast to these baselines, our approach creates pseudo categories based on spurious attributes (16-shot + 16-shot negative). For fairness, we ensure that the amount of training data is identical between the two baselines and our approach. We consider three typical methods for comparison, including CoCoOp [B], PromptSRC [C], and MaPLe [D], and evaluate them on the base-to-new generalization task as illustrated in the main paper. All results are averaged across 11 datasets.
>
> | Training Data                                 |   CoCoOp  |   MaPLe   | PromptSRC |  Average  |
> |-----------------------------------------------|:---------:|:---------:|:---------:|:---------:|
> | 16-shot main                                  |   70.05   |   75.39   |   73.78   |   72.94   |
> | 32-shot main                                  |   71.12   |   76.37   |   75.52   |   74.34   |
> | 16-shot main + 16-shot positive            |   70.34   |   75.80   |   74.46   |   73.53   |
> | 16-shot main + 16-shot negative (ours) | **73.50** | **77.69** | **77.88** | **76.36** |
>
> As shown in the table above, generating additional data using spurious attributes, i.e., negative samples, significantly outperforms the vanilla data augmentation, i.e., positive samples (76.36% vs 73.53%). Moreover, our proposed method even surpasses the performance of the 32-shot main (76.36% vs 74.34%). It is important to note that this comparison is not entirely fair to our method, as the latter relies on more labeled data from the original training set. This further suggests that the performance gains are primarily driven by the model's enhanced robustness to spurious attributes, rather than merely the augmented data. We have revised the paper and included this ablation experiment in Section B.11 of the Appendix (highlighted in blue).
>
> >**Q1: More visualization examples**
>
> Please refer to W2 above.
>
> >**Q2: Ablation study on performnace gains**
>
> Please refer to W3 above.
>
> **Reference**
>
> [D] MaPLe: Multi-modal Prompt Learning, CVPR2023

---

> > ### Comment · Reviewer_rex7 · 2024-11-25
> > **Response to authors**
> >
> > Thank you for your detailed response, which has addressed most of my concerns. I decide to maintain my rating.

---

> > > ### Author Response · Authors · 2024-11-25
> > > **Thank you for your feedback**
> > >
> > > Dear reviewer rex7,
> > >
> > > We greatly appreciate your efforts in providing valuable and insightful suggestions on this work. Your recognition of our article is highly significant to us.
> > >
> > > Thank you once again for your time.
> > >
> > > Best regards,
> > >
> > > Authors

---

> ### Author Response · Authors · 2024-11-25
> **Awaiting Your Valuable Feedback**
>
> Dear reviewer rex7,
>
> We sincerely appreciate your insightful comments and recognition of our work. We believe your feedback has been very helpful in enhancing the completeness and soundness of the paper. We have carefully addressed each concern and question with detailed responses and revisions. As the discussion deadline nears, we would like to confirm that if our revisions fully address your concerns. Please feel free to share any further questions or suggestions.
>
> Best,
>
> Authors

---

### Official Review · Reviewer_Yv7Q · 2024-10-31

**Soundness:** 3
**Presentation:** 3
**Contribution:** 3
**Rating:** 6
**Confidence:** 5

**Summary:**

This paper investigates how spuriously correlated attributes can lead to poor generalization in Vision-Language Models (VLMs). To address this issue, the authors propose the SAP method for identifying and filtering out problematic attributes, thereby enhancing existing attribute-based methods. They also introduce a plug-and-play SAS module to mitigate these attributes' influence on prediction. Experiments are conducted on standard benchmark datasets.

**Strengths:**

1. Researching spuriously correlated attributes in VLMs is a less explored yet intriguing direction.
2. The method of identifying spuriously correlated attributes is both reasonable and effective.
3. The SAS module can be integrated with existing methods to validate its generalization capability.

**Weaknesses:**

1. Most existing attribute-based methods are zero-shot, whereas the attribute filtering in this paper appears to require a few labeled training images. This raises concerns about whether comparing it to zero-shot attribute-based methods is fair.
2. Some of the latest methods in VLM adaptation have not been discussed. For example, [1,2], which explore VLM adaptation tasks from new perspectives, should be included in the discussion. It would be beneficial to verify the complementarity of the proposed method with these new approaches.
[1] MMA: Multi-Modal Adapter for Vision-Language Models, CVPR2024
[2] Dual memory networks: A versatile adaptation approach for vision-language models, CVPR2024

**Questions:**

See weaknesses.

---

> ### Author Response · Authors · 2024-11-19
> **Response to Reviewer Yv7Q by Authors**
>
> We greatly appreciate your insightful comments and your recognition of our work. Here are our responses to your concerns.
>
>
> >**W1: The comparison fairness to selected baselines**
>
>
> In Section 4 of the main paper, the baselines we use include prompt tuning, e.g., CoCoOp [A] and KgCoOp [B], attribute-based methods, e.g., ArGue [C] and CPL [D], as well as adapters [E], all of which are few-shot approaches that require a few labeled training data. Our proposed SAS, as a plug-and-play method, is consistent with them and uses the same training data. The only zero-shot baseline we compare against is zero-shot CLIP, as illustrated in Figure 3 of the main paper, which serves to indicate the vanilla performance of CLIP.
>
> >**W2: Evaluation of the proposed method on two new approaches**
>
> We thank the reviewer for the valuable suggestion. Here, as mentioned by the reviewer, we evaluate our method on the two recently proposed works. Specifically, for MMA [F], we train the newly introduced adapters in the deep layers that bridge the text and image representations, following their setting and implementation. For DMN [G], we optimize its memory projection functions and incorporate both the static and dynamic memory networks, which is the strongest variant according to their paper. We select the base-to-new generalization task, as illustrated in Section 4 of our main paper, and record the new category accurarcy, which directly reflects the generalization performance.
>
> | Method    | ImageNet  | Flowers102 | SUN397    | FGVCAircraft | StanfordCars |
> |-----------|-----------|------------|-----------|--------------|--------------|
> | MMA       |   71.00   |    75.93   |   78.57   |     36.33    |     73.10    |
> | MMA + SAS |   72.61   |    77.27   | **80.19** |   **37.85**  |     75.46    |
> | DMN       |   72.28   |    78.49   |   77.32   |     32.60    |     74.22    |
> | DMN + SAS | **73.34** |  **80.17** |   79.74   |     35.38    |   **76.30**  |
>
> As shown in the table above, SAS consistently improves performance on both methods, demonstrating its complementarity to the suggested approaches. In the revised paper, we have cited the mentioned works and included this experiment in Section B.16 of the Appendix.
>
> **Reference**
>
> [A] Conditional Prompt Learning for Vision-Language Models, CVPR2022
>
> [B] Visual-Language Prompt Tuning with Knowledge-guided Context Optimization, CVPR2023
>
> [C] ArGue: Attribute-Guided Prompt Tuning for Vision-Language Models, CVPR2024
>
> [D] Concept-Guided Prompt Learning for Generalization in Vision-Language Models, AAAI2024
>
> [E] CLIP-Adapter: Better Vision-Language Models with Feature Adapters, ArXiv
>
> [F] MMA: Multi-Modal Adapter for Vision-Language Models, CVPR2024
>
> [G] Dual Memory Networks: A Versatile Adaptation Approach for Vision-Language Models, CVPR2024

---

> ### Author Response · Authors · 2024-11-25
> **Awaiting Your Valuable Feedback**
>
> Dear reviewer Yv7Q,
>
> We would like to once again thank you for your insightful feedback and recognition of our work. We believe your comments have been instrumental in improving the comprehensiveness and clarity of our paper. We have presented responses and revisions addressing each listed concern and question. As the discussion deadline approaches, we would like to know if our revisions effectively address your concerns. Please let us know if you have any further questions or suggestions.
>
> Best,
>
> Authors

---

> > ### Comment · Reviewer_Yv7Q · 2024-11-27
> >
> > Thank you for your clarification. I will maintain my current positive rating.

---

> ### Author Response · Authors · 2024-11-27
> **Thank you for your feedback**
>
> Dear reviewer Yv7Q,
> ​
>
> ​We sincerely appreciate your response and recognition of our work.
> ​
>
> ​Thank you once again for your time on reviewing this paper.
> ​
>
> ​Best regards,
> ​
>
> ​Authors

---

### Official Review · Reviewer_9qjN · 2024-11-02

**Soundness:** 3
**Presentation:** 3
**Contribution:** 2
**Rating:** 8
**Confidence:** 3

**Summary:**

This paper aims to address the issue of VLMs overly relying on a small subset of attributes in decision-making due to spuriously correlated attributes. The paper proposes SAP to identify and filter out spurious attributes to enhance the generalization of existing attribute-based methods. In addition, a plug-and-play module SAS that integrates into various Parameter-Efficient Fine-Tuning (PEFT) methods is designed to reduce the influence of spurious attributes on predictions. The experiments demonstrate that SAP and SAS significantly improve accuracy on distribution shifts across 11 datasets and 3 generalization tasks without compromising downstream performance.

**Strengths:**

- Novelty: The paper reveals the impact of spuriously correlated attributes on the predictions of Vision-Language Models and proposes innovative methods (SAP and SAS) to tackle the problem of spurious correlations in VLMs

- Performance & Experiments: The proposed methods achieve state-of-the-art results across 11 datasets and 3 generalization tasks, enhancing the accuracy of VLMs on distribution shifts without sacrificing performance on downstream tasks. The figures of results effectively express the intended meaning and support the conclusions, as illustrated in Figure 3.

- Presentation: The writing is fluent and understandable.

**Weaknesses:**

- Does the construction of pseudo categories using the synthetic method result in additional computational costs? And have any experiments been conducted to validate the computational efficiency of this method? Additionally, could you provide some analysis of the inference time based on GPT and diffusion-related methods?
- The paper primarily focuses on vision-language recognition. Could the proposed method generalize to other modalities (such as video) or tasks (such as language reasoning) outside of this specific task in this paper?

**Questions:**

- Will the code be made publicly available for community use?

---

> ### Author Response · Authors · 2024-11-19
> **Response to Reviewer 9qjN by Authors (1/2)**
>
> We are grateful for your valuable comments and your acknowledgment of our work. Below are our responses to your concerns.
>
> >**W1: The computational efficiency of the proposed method**
>
> As suggested by the reviewer, here we provide a detailed overview of the computation and time costs of the proposed method.
>
> - **The cost of training.** Since the construction of pseudo categories augments the original data, the training process may need more computation to converge. However, we want to emphasize that our method introduces additional computations only during training, without adding any extra computations during testing. Furthermore, to alleviate the training computation, in Section 4 of the main paper, we introduce selective optimization trick. Instead of optimizing every main category, we focus on a small subset of categories that seriously suffer from spurious correlations. As demonstrated in Table 5 of the main paper, this approach significantly reduces training time while preserving the majority of the accuracy gains.
>
> - **The cost of diffusion generation.** Here, we provide the estimated inference time required to construct pseudo categories through Stable Diffusion for each dataset. Please refer to Section A.4 of the Appendix for the detailed settings and prompts of the diffusion.
>
> | Dataset      | Inference Time |
> |--------------|:--------------:|
> | Caltech101   |      25min     |
> | OxfordPets   |      10min     |
> | StanfordCars |      45min     |
> | Flowers102   |      30min     |
> | Food101      |      25min     |
> | FGVCAircraft |      30min     |
> | SUN397       |      90min     |
> | DTD          |      15min     |
> | EuroSAT      |      5min      |
> | UCF101       |      25min     |
> | ImageNet     |     3h50min    |
>
>
> Note that due to time constraints, the inference time provided above is an approximation based on the time required for each generation. As shown, the total inference time is proportional to the size of the dataset, particularly the number of categories involved. For most datasets, the inference time is under half an hour, and the entire inference process can be completed within half a day. It is important to note that this is a one-time operation, and no additional inference is needed during subsequent training.
>
> - **The cost of GPT prompting.** In our method, a key step is identifying the spurious attributes within each category, which we accomplish by prompting MLLMs, i.e., GPT. As suggested by the reviewer, we provide the time cost of this process along with a thorough analysis. Specifically, to enhance efficiency, we employ batch inference as implemented in [A], where multiple queries can be processed concurrently, which significantly reduces the inference time for GPT.
>
> | Dataset      | Inference Time |
> |--------------|:--------------:|
> | Caltech101   |      10min      |
> | OxfordPets   |      10min      |
> | StanfordCars |      25min     |
> | Flowers102   |      10min      |
> | Food101      |      10min      |
> | FGVCAircraft |      10min      |
> | SUN397       |      35min     |
> | DTD          |      5min      |
> | EuroSAT      |      3min      |
> | UCF101       |      10min      |
> | ImageNet     |      1h30min     |
>
> As shown in the table above, the GPT inference time for most datasets is under 10 minutes. The complete inference process takes approximately three hours, which is also a one-time operation that does not need to be repeated thereafter. It is worth noting that upon obtaining the responses, we need to perform post-processing such as filtering and selection to determine valid attributes, as detailed in Section A.3 of the Appendix, which may require additional time.
>
> We have included the above statistics and analysis regarding computation and time costs in Section B.14 of the revised paper. (highlighted in blue)
>
> **Reference**
>
> [A] Visual Classification via Description from Large Language Models, ICLR2023

---

> > ### Author Response · Authors · 2024-11-19
> > **Response to Reviewer 9qjN by Authors (2/2)**
> >
> > >**W2: Evaluation on other modalities or tasks**
> >
> > To assess the transferability of our method to other modalities or tasks, we explore video recognition and leave more tasks, such as language reasoning, for future work. Specifically, we choose ViFi-CLIP [B], a fully fine-tuned CLIP model tailored for video understanding. ViFi-CLIP employs a training framework similar to CLIP, incorporating a temporal pooling layer to derive video representations from multiple frames. Following the base-to-new generalization setting in [B], we evaluate video-level generalization performance on four video datasets: K-400 [C], HMDB-51 [D], UCF-101 [E], and SSv2 [F]. As in the main paper, we select three representative baseline methods: CoCoOp [G], MaPLe [H], and PromptSRC [I]. Since ViFi-CLIP shares its architecture with CLIP, these methods can be easily transferred to ViFi-CLIP, which has been implemented by [I]. We incorporate the proposed method, SAS, into these baselines to verify its effectiveness by contrasting spurious attributes with each frame of the video. We record the new category accuracy for selected datasets which directly reflects the generalization performance on unseen categories.
> >
> > | Method          |   K-400   |  HMDB-51  |  UCF-101  |    SSv2   |
> > |-----------------|:---------:|:---------:|:---------:|:---------:|
> > | ViFiCLIP        |   61.10   |   53.30   |   67.70   |   12.10   |
> > | CoCoOp          |   64.70   |   54.41   |   68.21   |   14.24   |
> > | CoCoOp + SAS    |   66.39   |   56.64   |   70.40   |   16.01   |
> > | MaPLe           |   64.52   |   58.23   |   70.73   |   14.74   |
> > | MaPLe + SAS     |   66.42   |   59.32   |   72.66   |   16.40   |
> > | PromptSRC       |   68.31   |   62.38   |   76.79   |   17.22   |
> > | PromptSRC + SAS | **70.23** | **64.70** | **79.31** | **18.95** |
> >
> > As shown in the table above, despite the input modalities shifting from images to videos, SAS consistently delivers performance gains across all datasets, proving it to be an effective plug-and-play method that can be generalized to more complex modalities and tasks. In the revised paper, we have added this experiment to Section B.15 of the Appendix.
> >
> > >**Q1: The availability of code for community use**
> >
> > We will release the code for implementation and the corresponding technical documents upon the publication of our paper.
> >
> > **Reference**
> >
> > [B] Fine-tuned CLIP Models are Efficient Video Learners, CVPR2023
> >
> > [C] The Kinetics Human Action Video Dataset, ArXiv
> >
> > [D] HMDB: A large video database for human motion recognition, ICCV2011
> >
> > [E] UCF101: A Dataset of 101 Human Actions Classes From Videos in The Wild, ArXiv
> >
> > [F] The "something something" video database for learning and evaluating visual common sense, CVPR2017
> >
> > [G] Conditional Prompt Learning for Vision-Language Models, CVPR2022
> >
> > [H] MaPLe: Multi-modal Prompt Learning, CVPR2023
> >
> > [I] Self-regulating Prompts: Foundational Model Adaptation without Forgetting, ICCV2023

---

> > > ### Comment · Reviewer_9qjN · 2024-11-24
> > > **Rebuttal Response**
> > >
> > > Thank you for your detailed responses, and I especially appreciate the additional experimental results in the short time frame.
> > > Your response has addressed most of my concerns.
> > >
> > > However, I still have one additional question to confirm and one concern to address:
> > >
> > > - Question: Is SAS only applied during the training phase, with the testing phase consistent with the original model w/o SAS?
> > > - Concern: Based on the experimental results you provided regarding GPT and SD, I remain concerned about the additional training overhead introduced by incorporating GPT and SD. （It doesn’t diminish my recognition of this paper’s contribution to improving model performance.）
> > >
> > > I will consider maintaining or even increasing my score.

---

> ### Author Response · Authors · 2024-11-24
> **Follow-up Response by Authors**
>
> We would like to sincerely express our gratitude once again for your response and positive feedback on our work. Below, we have provided further responses to your question and concern.
>
> >**Q: Is SAS deployed only in training?**
>
> The short answer is yes. The essence of SAS lies in incorporating an auxiliary objective during training, enabling the model to distinguish between main objects and spurious attributes, thus enhancing its robustness to spurious features. Unlike adapters or prompt tuning, SAS does not introduce additional learnable parameters. Therefore, during testing, the model remains consistent with the original, without adding computational overhead.
>
> >**W: The training overhead by incorporating GPT and SD**
>
> We understand your concerns. Here, we further elaborate and clarify the efficiency of model training and the inference of GPT or SD.
>
> **Regarding model training**, in Section B.14 of the revised paper, we have included the training time statistics of SAS on additional datasets beyond ImageNet as shown in Table 14. For your convenience, we have directly displayed the table below.
>
> | Method             | Flowers102 | Food101 | FGVCAircraft | StanfordCars | Average Time |
> |--------------------|:----------:|:------:|:------------:|:------------:|:------------:|
> | CoCoOp             |   10m48s   | 18m34s |     8m02s    |    12m46s    |    12m32s    |
> | + SAS              |   13m14s   | 24m07s |    13m36s    |    17m08s    |    17m01s    |
> | + selective trick  |   11m03s   | 20m45s |    10m23s    |    14m55s    |    14m16s    |
> | PromptSRC          |    6m25s   | 15m09s |     5m44s    |     9m36s    |     9m13s    |
> | + SAS              |    8m26s   | 18m21s |     7m11s    |    12m04s    |    11m30s    |
> |  + selective trick |    6m58s   | 16m22s |     6m35s    |    10m20s    |    10m03s    |
>
> As shown in the table above, for most datasets, the integration of SAS only increases the training time by approximately 3 to 5 minutes, while selective optimization further reduces this time to a negligible amount. In fact, the selective optimization trick is proposed to address large-scale datasets, such as ImageNet, which contains 1000 categories. For regular datasets (~100 categories), the time consumption of SAS is fully acceptable.
>
> **Regarding inference of GPT and SD**, for the former, we would like to emphasize that, compared to conventional works that identify visual attributes via manual labeling [J, K] as introduced in Section 2, directly obtaining spurious attributes via GPT is a much more efficient and cost-effective method. Moreover, following the implementation of [A], we have limited the prompting time for most datasets to within 10 minutes.
>
> For the latter, since we employ standard diffusion hyperparameters following [L], we believe that by exploring more settings, such as reducing the number of diffusion steps or deploying different schedulers, we may achieve a better trade-off between accuracy and efficiency. Due to time constraints, we leave this as future work. Nonetheless, we want to clarify that, for most datasets, the SD inference time can still be kept under half an hour. It is worth mentioning that throughout the experiment, we use a single NVIDIA 4090 GPU as described in Section 4, and we expect that utilizing more or more powerful GPUs would further reduce the required time.
>
> Thank you again for your recognition of our work. We believe your comments have greatly improved the comprehensiveness and soundness of the paper. We hope that our further responses may address your remaining concerns.
>
> **Reference**
>
> [J] How to discover spurious features in deep learning?, ICLR2021
>
> [K] Leveraging sparse linear layers for debuggable deep networks, ICML2021
>
> [L] SuS-X: Training-Free Name-Only Transfer of Vision-Language Models, ICCV2023

---

> ### Author Response · Authors · 2024-11-24
> **Follow-up Response by Authors (Update)**
>
> To better address the reviewer's efficiency concern, here we update the response with a simple ablation study on the efficiency of diffusion inference. Specifically, we vary the number of diffusion steps, which is the key hyperparameter determining the inference time cost. Intuitively, fewer steps are more efficient yet yield lower image quality, while more steps ensure image fidelity but require more computation. We select CoCoOp as the baseline and record the new category accuracy on base-to-new generalization. Due to time constraints, here we provide the results for four representative datasets.
>
> | Step |     Flowers102    |      Food101      |    FGVCAircraft   |    StanfordCars   |    Average   |
> |------|:-----------------:|:-----------------:|:-----------------:|:-----------------:|:-----------------:|
> |  25  |   72.24 (10min)   |    91.18 (6min)   |    27.23 (8min)   |   73.55 (13min)   |    66.05 (9min)   |
> |  50  |   72.85 (15min)   |   91.96 (11min)   | **28.41** (14min) |   74.67 (23min)   |   66.97 (16min)    |
> |  75  |   72.81 (22min)   | **92.28** (18min) |   28.19 (24min)   |   74.82 (31min)   |   67.02 (24min)   |
> |  100 (default) | **72.99** (28min) |   92.12 (26min)   |   28.30 (31min)   | **74.96** (40min) | **67.09** (31min) |
>
> As shown in the table above, by default, we use 100 steps throughout the paper as described in Section A.4, which requires an average of 31 minutes to generate images per dataset. Here we try fewer steps, such as 50, and observe that the time required for diffusion nearly halves (31min -> 16min) with minimal degradation in performance (67.09 -> 66.97). However, while the number of steps is further reduced to 25, there is a dramatic performance drop (66.97 -> 66.05), possibly due to the decline in image quality. This suggests we may safely adjust the number of steps from 100 to 50, which halves the required time with minimal accuracy loss, significantly improving the efficiency of SAS.
>
> This simple experiment highlights SAS's significant potential for improved efficiency. With advancements in diffusion sampling strategies, we believe that the computational overhead introduced by diffusion will soon no longer be a concern. We have added this ablation study in Section B.17 of the revised paper. Further attempts with different settings, such as various schedulers or checkpoints, are left as future work.

---

> > ### Comment · Reviewer_9qjN · 2024-11-27
> > **Rebuttal Response**
> >
> > Thank you for your response. I will increase my rating to 8.

---

> > > ### Author Response · Authors · 2024-11-27
> > > **Thank you for your feedback**
> > >
> > > Dear reviewer 9qjN,
> > >
> > > We sincerely appreciate your positive feedback. Your recognition of this work means a lot to us.
> > >
> > > Thank you once again for your time on reviewing the paper.
> > >
> > >
> > > Best,
> > >
> > > Authors

---

### Official Review · Reviewer_bxnj · 2024-11-02

**Soundness:** 2
**Presentation:** 2
**Contribution:** 2
**Rating:** 6
**Confidence:** 5

**Summary:**

1. [Summary]
This paper focuses on Parameter-Efficient Fine-Tuning for vision language models. It discovers a group of black sheep, i.e., spurious attributes, on which VLMs inherently heavily rely, thereby leading to poor generalization and robustness. It introduces SPURIOUS ATTRIBUTE PROBING (SAP), aiming to identify and eliminate these problematic attributes, thereby improving the generalization of current attribute-based methods. Besides, it presents SPURIOUS ATTRIBUTE SHIELDING (SAS), a plug-and-play module seamlessly integrating into various PEFT methods to mitigate the influence of spurious attributes on predictions. Experiments show that the proposed method achieves good performances.
2. [Strengths]
- a. It proposes SPURIOUS ATTRIBUTE PROBING (SAP) and SPURIOUS ATTRIBUTE SHIELDING (SAS), which enhance the generalization of existing attribute-based methods is complementary to PEFT methods.
- b. The experiments are extensive, and the performance of the proposed method is promising.
- c. The method is efficient, intuitive and effective.
3.1 [Reasons to Reject]
Although this paper is well written with comprehensive evaluation and good results, there are still some issues.
Several parts of this paper are not very clear and need further clarification. Please check the questions.
In addition, some key related VLM works are missed. Please check the questions.
3. [Weaknesses]
1. Fair comparison issue. It is not clear whether the performance gains come from involving additional training data as illustrated in Figure 2 (a) and (b) or the designed methods. What if the compared methods also involve the additional data in training? And how about the performance then? It seems that the performance gains are largely attributed to the additional training data. It would be better to add some additional ablation studies to address this concern.
2. The core idea of this paper is to introduce attribute information (e.g., objects, parts and subparts) into vision-language model to enhance object recognition. The related works [A, B, C] are missed. Similarly, [A, B, C] enhance object recognition performance by introducing the additional information like multi-granularity object class information and hierarchical object class information. It would be better to discuss and highlight the similarity and difference between this paper and [A, B, C].
[A] Adversarial fine-grained composition learning for unseen attribute-object recognition
[B] Open-Vocabulary Object Detection via Language Hierarchy
[C] Multiple granularity descriptors for fine-grained categorization

3. The paper method relies on constructing pseudo categories via generating or retrieving, which shares similar idea with the concept of retrieval-augmented generation (RAG). It would be better to discuss and clarify the similarity and differences with RAG, for example, RAG for object recognition as in [D].
[D] Retrieval augmented classification for long-tail visual recognition.
4. As the objective of this paper is to enhance vision-language models, it would be better to test/verify the proposed method on multiple vision-language models in addition to CLIP. Besides, the Section 2 Related Work missing an important paragraph of “Vision-language model.”. It would be better to add a new paragraph in the Section 2 Related Work to introduce background and related papers of VLM and cite the survey [E] which could be a good reference for new readers that are not very familiar with recent progresses in VLM.
[E] Vision-language models for vision tasks: A survey


4. Conclusion
Overall, this work proposes to introduce attribute information into VLM to enhance object recognition performance with good performance gains. However, there are some details that need to be clarified, as listed in the questions. I would upgrade the score if the questions are well addressed.

**Strengths:**

as shown in Summary

**Weaknesses:**

as shown in Summary

**Questions:**

as shown in Summary

---

> ### Author Response · Authors · 2024-11-12
> **Missing reference in the review**
>
> Thank you for the insightful comments. The reference section for [A]-[E] is not displayed in the review. Could you update that part?

---

> ### Comment · Reviewer_bxnj · 2024-11-13
> **Please check the review carefully.**
>
> Please check the review carefully. Reference [A-C] is provided at the end of Weaknesses 5 .  Reference [D] is provided at the end of Weaknesses 6 .  Reference [E] is provided at the end of Weaknesses 7 .

---

> ### Author Response · Authors · 2024-11-19
> **Response to Reviewer bxnj by Authors (1/3)**
>
> We would like to thank you once again for your constructive comments and questions. Following are our responses to your concerns.
>
> > **W1: Performance gains from extra data or the proposed method**
>
> SAS improves model generalization by training it to differentiate between main objects and spurious attributes through the creation of pseudo categories that feature the latter. Therefore, the performance gains should result from enhanced model robustness to spurious concepts instead of the vanilla constructed data. In Section 4.2 of our main paper, we perform an ablation study to confirm the contribution of spurious attributes on our method by adjusting $\gamma$.
>
> Here, as suggested by the reviewer, we further compare the performance gains brought by extra data and our proposed method to validate this point. Specifically, in addition to the proposed method, we design two baselines. In the first baseline, we consider additional data directly from the original dataset featuring the main objects, where we extend the training data from 16 shots to 32 shots (32-shot main).  In the second baseline, we involve additional data generated by pseudo categories, where instead of featuring spurious attributes, these pseudo categories are the same as the main categories, i.e., vanilla constructed data (16-shot main + 16-shot pseudo main). In contrast to the first two baselines, our approach creates pseudo categories based on spurious attributes (16-shot main + 16-shot pseudo spurious). For fairness, we ensure that the amount of training data is identical between the two baselines and our approach. We select three typical methods for comparison, including CoCoOp [A], MaPLe [B], and PromptSRC [C], and evaluate them on the base-to-new generalization task. All results are averaged across 11 datasets, as illustrated in the main paper.
>
> | Training Data                                 |   CoCoOp [A]  |   MaPLe [B]   | PromptSRC [C] |  Average  |
> |-----------------------------------------------|:---------:|:---------:|:---------:|:---------:|
> | 16-shot main                                  |   70.05   |   75.39   |   73.78   |   72.94   |
> | 32-shot main                                  |   71.12   |   76.37   |   75.52   |   74.34   |
> | 16-shot main + 16-shot pseudo main            |   70.34   |   75.80   |   74.46   |   73.53   |
> | 16-shot main + 16-shot pseudo spurious (ours) | **73.50** | **77.69** | **77.88** | **76.36** |
>
>
> As shown in the table above, generating additional data using spurious attributes significantly outperforms vanilla constructed data for main categories (76.36% vs 73.53%). Furthermore, our proposed method even exceeds the performance of the 32-shot main (76.36% vs 74.34%). It is important to note that this comparison is not entirely fair for our method, as the latter relies on more labeled data from the original training set. This further suggests that the performance gains are primarily driven by the model's enhanced robustness to spurious attributes, rather than merely the increased training data. We have revised the paper and included this ablation experiment in Section B.11 of the Appendix (highlighted in blue).
>
> **Reference**
>
> [A] Conditional Prompt Learning for Vision-Language Models, CVPR2022
>
> [B] MaPLe: Multi-modal Prompt Learning, CVPR2023
>
> [C] Self-regulating Prompts: Foundational Model Adaptation without Forgetting, ICCV2023

---

> > ### Author Response · Authors · 2024-11-19
> > **Response to Reviewer bxnj by Authors (2/3)**
> >
> > >**W2: Discussion on suggested related works**
> >
> > In Section 2 of the main paper, we provide an overview of recent works related to visual attributes, and furthermore, we introduce similar works on spurious attribute identification and mitigation for model robustness and generalization. We appreciate the reviewer for pointing out additional related works that we may need to discuss. Here, we highlight the differences and connections between our work and [D, E, F].
> >
> > - **From a motivational perspective, the categories of attributes we discuss are different.** The related works mentioned primarily improve object recognition by introducing core attributes, which are typically properties or components of the main object. For example, "young tiger" demonstrated in [D], where "young" is an attribute featuring "tiger" itself. The attributes used in [E, F], derived from WordNet, serve as alternative descriptions for main objects. In contrast, our work focuses on spurious attributes, which often occur alongside main objects but are not inherently part of them. For instance, "tiger" commonly co-occurs with "forest", with the latter considered a spurious attribute. In Section 1 of the main paper, we also conceptually introduce the distinction between core attributes and spurious attributes. Unlike core attributes, which aid in recognition by fine-grained descriptions, spurious attributes tend to lead models to learn spurious correlations, e.g., associating "tiger" with "forest", resulting in poor generalization while encountering out-of-distribution datasets.
> > - **From a methodological perspective, the way we utilize attributes are different.** The two distinct categories of attributes mentioned above result in fundamental differences in our methods. For instance, [D] employs adversarial training to learn the composition between core attributes and objects, while [E, F] introduce multi-granularity and hierarchical descriptions to enhance the model's discrimination capability. These approaches implicitly reinforce the association between core attributes and main objects, thus improving fine-grained visual  understanding. In contrast, our proposed method, SAS, emphasizes the distinction between main objects and spurious attributes, aiming to enhance the model's ability to differentiate between them and reduce reliance on spurious attributes, thereby improving robustness.
> >
> > - **From a data-centric perspective, the attribute source we employ are different.** For example, [D] uses existing object-attribute pair datasets for training, while [E, F] leverages WordNet's ontology tree to find relevant visual descriptions. However, for some uncommon objects that do not appear in previous datasets or WordNet, there are no available attributes. In contrast, our proposed method, SAP, derives the corresponding core attributes and spurious attributes for arbitrary objects through MLLM prompting, without being limited by existing data sources.
> >
> > We have cited and introduced the mentioned works in Section 2 of the revised paper (highlighted in blue).
> >
> > **Reference**
> >
> > [D]  Adversarial fine-grained composition learning for unseen attribute-object recognition, ICCV2019
> >
> > [E] Open-Vocabulary Object Detection via Language Hierarchy, NeurIPS2024
> >
> > [F] Multiple granularity descriptors for fine-grained categorization, ICCV2015

---

> > > ### Author Response · Authors · 2024-11-19
> > > **Response to Reviewer bxnj by Authors (3/3)**
> > >
> > > >**W3: Relationship between our work and RAG**
> > >
> > > We agree with the reviewer that our work shares some motivations with RAG. Here, as suggested by the reviewer, we clarify the connections and distinctions between our work and RAG.
> > >
> > > - **The connection to RAG.** RAG is proposed essentially to address the insufficiency or lack of desired data. For example, [G], as mentioned by the reviewer, improves long-tail recognition performance by retrieving text representations for tail classes. Similarly, [H] enhances VLMs' tail accuracy by identifying and retrieving high-frequency text synonyms corresponding to tail names from the training set. Furthermore, [I] mitigates data sparsity issues by retrieving external images through class names for data augmentation. In line with previous work, we construct pseudo categories featuring spurious attributes through retrieval, thereby enhancing the model's robustness to these attributes.
> > >
> > > - **The distinction to RAG.** Beyond retrieval, we also explore data synthesis. In Section B.5 of the Appendix, we compare the performance of our method using synthesized and retrieved data, empirically concluding that synthesized data yields greater accuracy gains. Compared to retrieval, synthesis can offer more tailored and precise scenarios and objects, which may be more suitable for our method given the diverse identified attributes.
> > >
> > > In the revised paper, we have cited the suggested work and included the details of RAG and its relationship to our work in Section C.6 of the Appendix (highlighted in blue).
> > >
> > > >**W4: Evaluation on other VLMs**
> > >
> > > We appreciate the reviewer's suggestion. Following previous works [A, B, C], we select CLIP as a representative VLM to evaluate the effectiveness of our method in the main paper. Here, as suggested by the reviewer, we test the proposed method on additional VLMs, including BLIP [J], CLIPA-v2 [K], EVA-CLIP [L], and SigLIP [M]. Our experimental settings remain consistent with that mentioned above.
> > >
> > > | Model          |   CoCoOp  |   MaPLe   | PromptSRC |  Average  |
> > > |----------------|:---------:|:---------:|:---------:|:---------:|
> > > | BLIP           |   68.81   |   72.32   |   72.62   |   70.58   |
> > > | BLIP + SAS     |   70.54   |   74.35   |   73.97   |   72.95   |
> > > | CLIPA-v2       |   70.28   |   73.40   |   74.52   |   72.73   |
> > > | CLIPA-v2 + SAS |   72.42   |   74.88   |   77.08   |   74.79   |
> > > | EVA-CLIP       |   72.75   |   77.58   |   76.13   |   75.49   |
> > > | EVA-CLIP + SAS |   74.60   | **77.92** |   77.82   |   76.78   |
> > > | SigLIP         |   74.99   |   73.78   |   78.64   |   75.80   |
> > > | SigLIP + SAS   | **76.41** |   75.26   | **79.87** | **77.18** |
> > >
> > > As demonstrated in the table above, our proposed method, SAS, consistently yields performance gains across a range of VLMs, extending beyond just CLIP. We have included this experiment in Section B.13 of the revised paper (highlighted in blue).
> > >
> > > >**W5: Introduction of background and related works of VLMs**
> > >
> > > In Section 1 of the main paper, we provide a brief introduction of VLMs and the recently widely discussed generalization issue, which motivates our study. We agree that a more detailed background on VLMs, or including more surveys of VLMs such as [N] suggested by the reviewer, could give readers a clearer understanding of the recent progress of VLMs, making it easier to grasp the insights of this work. In the revised paper, we have added a separate paragraph in Section 2 which provides an introduction of VLMs and refer to the mentioned survey for readers (highlighted in blue).
> > >
> > > **Reference**
> > >
> > > [G] Retrieval augmented classification for long-tail visual recognition, CVPR2022
> > >
> > > [H] The Neglected Tails in Vision-Language Models, CVPR2024
> > >
> > > [I] SuS-X: Training-Free Name-Only Transfer of Vision-Language Models, ICCV2023
> > >
> > > [J] BLIP: Bootstrapping Language-Image Pre-training for Unified Vision-Language Understanding and Generation, ICML2022
> > >
> > > [K] An Inverse Scaling Law for CLIP Training, NeurIPS2023
> > >
> > > [L] EVA-CLIP: Improved Training Techniques for CLIP at Scale, ArXiv
> > >
> > > [M] Sigmoid Loss for Language Image Pre-Training, ICCV2023
> > >
> > > [N] Vision-Language Models for Vision Tasks: A Survey, TPAMI

---

> ### Author Response · Authors · 2024-11-25
> **Awaiting Your Valuable Feedback**
>
> Dear reviewer bxnj,
>
> We sincerely appreciate the effort you dedicated to reviewing our paper. Your feedback has greatly enhanced the solidity and clarity of our work. We have provided thorough responses and revisions based on your concerns and suggestions. As the discussion end date approaches, we are eager to know if our revisions have addressed your comments, and we welcome any further questions or suggestions.
>
> Best,
>
> Authors

---

> > ### Comment · Reviewer_bxnj · 2024-11-27
> >
> > Thank you for your responses, which have addressed most of my concerns. I would like to raise my rating.

---

> > > ### Author Response · Authors · 2024-11-27
> > > **Thank You for Your Feedback**
> > >
> > > Dear reviewer bxnj,
> > >
> > > We are sincerely grateful for your response and happy to know that we effectively address your concerns.
> > >
> > > Thank you once again for your time on reviewing this paper.
> > >
> > > Best regards,
> > >
> > > Authors

---

### Author Response · Authors · 2024-11-22
**Summary of Revision**

Dear reviewers and AC,

We are sincerely grateful for the valuable time you dedicated to reviewing our paper and offering insightful feedbacks. We have carefully revised the paper in line with your comments and provided comprehensive responses to all your inquiries. Specifically,

1. Added an ablation study on performance gains in Section B.11 (reviewer bxnj & rex7)
2. Included discussions on more related works and background of VLMs in Section 2 and Section C.6 (reviewer bxnj)
3. Conducted evaluations on additional VLMs and methods in Section B.13 and Section B.16 (reviewer bxnj & Yv7Q)
4. Exploring applications on video tasks in Section B.15 (reviewer 9qjN)
5. Provided statistics and analysis on efficiency of the proposed method in Section B.14 (reviewer 9qjN & rex7)
6. Added more visualization examples in Section B.12 (reviewer rex7)

We genuinely hope these revisions have adequately addressed your concerns. As the **discussion deadline of November 27** is approaching, please feel free to reach out if you have any further concerns or suggestions.

Best,

Authors

---

### Author Response · Authors · 2024-11-24
**[Gentle reminder] three days remaining before discussion deadline**

As the discussion deadline (Nov 26, UTC-12) draws near, we kindly invite all reviewers to check out the revised version of the paper and Appendix. We have provided detailed, point-by-point responses to each concern and question. We look forward to hearing whether our revisions have addressed your comments and are open to any further questions or suggestions.

---

### Meta-Review · Area_Chair_SRqT · 2024-12-17

**Metareview:**

This paper proposes Spurious Attribute Probing (SAP) and Spurious Attribute Shielding (SAS) for few-shot adaptation for Vision-Language Models (VLMs). The paper is reviewed by four reviewers.

The strengths of the paper include: 1) two novel methods; 2) extensive experiments; 3) well-written; 4) strong performance; 5) the study of spuriously correlated attributes in VLMs is a less explored yet intriguing direction.

Initially, the reviewers have raised several concerns and drawbacks to this paper. The authors have provided a rebuttal. After checking the rebuttal and the comments of the other reviewers, the reviewers consistently agreed that the authors have solved their concerns and would like to recommend acceptance to this paper. Finally, this paper got two accept and two weak accept. The AC thinks this paper could be a valuable work for the community of few-shot adaption with VLM and would like to recommend acceptance to this paper.

**Additional Comments On Reviewer Discussion:**

Initially, the reviewers have raised several concerns and drawbacks to this paper. The authors have provided a rebuttal. After checking the rebuttal and the comments of the other reviewers, the reviewers consistently agreed that the authors have solved their concerns. Two reviewers have raised the scores, from 5 to 6 and 6 to 8, respectively.

---

### Decision · Program_Chairs · 2025-01-22

Accept (Poster)